# Multiple time-scales of decision-making in the hippocampus and prefrontal cortex

**Wenbo Tang[1†], Justin D Shin[1†], Shantanu P Jadhav[1,2]\***

[1]Graduate Program in Neuroscience, Brandeis University, Waltham, United States; [2]Neuroscience Program, Department of Psychology, and Volen National Center for Complex Systems, Brandeis University, Waltham, United States

**Abstract** The prefrontal cortex and hippocampus are crucial for memory-guided decision-making. Neural activity in the hippocampus exhibits place-cell sequences at multiple timescales, including slow behavioral sequences (~seconds) and fast theta sequences (~100–200 ms) within theta oscillation cycles. How prefrontal ensembles interact with hippocampal sequences to support decision-making is unclear. Here, we examined simultaneous hippocampal and prefrontal ensemble activity in rats during learning of a spatial working-memory decision task. We found clear theta sequences in prefrontal cortex, nested within its behavioral sequences. In both regions, behavioral sequences maintained representations of current choices during navigation. In contrast, hippocampal theta sequences encoded alternatives for deliberation and were coordinated with prefrontal theta sequences that predicted upcoming choices. During error trials, these representations were preserved to guide ongoing behavior, whereas replay sequences during inter-trial periods were impaired prior to navigation. These results establish cooperative interaction between hippocampal and prefrontal sequences at multiple timescales for memory-guided decision-making.

**\*For correspondence:**
shantanu@brandeis.edu

[†]These authors contributed equally to this work

**Competing interests:** The authors declare that no competing interests exist.

## Introduction

The neural substrates that support decision-making are still not fully understood. The link between decision-making and neural representations at the behavioral timescale has been studied extensively in various cortical and sub-cortical circuits of different species. Early classic work showed that during tasks involving sustained attention or decision-making, neurons in the prefrontal cortex (PFC) and the posterior parietal cortex (PPC) can exhibit persistent activity over seconds throughout retention intervals for maintenance of decision-related information (*Fuster, 2015*; *Goldman-Rakic, 1995*; *Miller et al., 2018*; *Sreenivasan and D'Esposito, 2019*). In contrast to these low-dimensional representations that require long-lived stable states, decision-related information can also be held in a dynamic population code. At the ensemble level, heterogenous activity patterns comprising sequences of neuronal activation that span entire task periods have emerged as a common coding scheme in many brain regions, including PFC (*Baeg et al., 2003*; *Fujisawa et al., 2008*; *Ito et al., 2015*), hippocampus (*Ito et al., 2015*; *Pastalkova et al., 2008*), PPC (*Crowe et al., 2010*; *Harvey et al., 2012*), and striatum (*Bakhurin et al., 2016*; *Barnes et al., 2005*).

In addition to this behavioral-timescale activity, recent work has raised the possibility that neural dynamics at fast, cognitive timescales that occur transiently during discrete subsets of task periods can also underlie upcoming decisions. In many brain areas, such as prefrontal, parietal, orbitofrontal cortices, and hippocampus, population activity can change in an abrupt, coordinated, and transient manner in support of flexible decisions (*Bernacchia et al., 2011*; *Durstewitz et al., 2010*; *Johnson and Redish, 2007*; *Karlsson et al., 2012*; *Latimer et al., 2015*; *Lundqvist et al., 2016*; *Rich and Wallis, 2016*). For example, discrete transient bursts of gamma and beta oscillations in PFC have been shown to increase with working-memory load during delays (*Lundqvist et al., 2016*).

In particular, recent studies have identified time-compressed neuronal sequences in the hippocampus as a specific cell-assembly pattern at fast timescales that can support decision-making processes.

Hippocampal theta sequences provide time-compressed ensemble representations of spatial paths within single cycles of theta oscillations (6–12 Hz) during active navigation, which reflect a candidate neural mechanism for planning at decision time (*Johnson and Redish, 2007*; *Kay et al., 2020*; *Papale et al., 2016*; *Pezzulo et al., 2019*; *Zheng et al., 2020*; *Zielinski et al., 2020*). Whether theta sequences exist in PFC during decision-making has yet to be determined. Further, during pauses in exploration, replay sequences are observed in the hippocampus during individual sharp-wave ripples (SWRs; *Carr et al., 2011*), and are known to interact with PFC to reactivate past and future trajectories on the temporal scale of 100–200 ms during memory-guided decisions (*Shin et al., 2019*; *Tang and Jadhav, 2019*). Disruption of fast sequences in the hippocampus, while leaving behavioral-timescale spatial representations intact, can impair navigation decisions (*Fernández-Ruiz et al., 2019*; *Jadhav et al., 2012*; *Petersen and Buzsáki, 2020*; *Robbe and Buzsáki, 2009*). Thus, fast hippocampal sequences are promising transient activity patterns that can support decision-making at sub-second speed (*Buzsáki et al., 2014*; *Kay et al., 2020*; *Papale et al., 2016*; *Pezzulo et al., 2019*; *Shin et al., 2019*). Whether and how these fast-timescale representations are organized together in hippocampal-prefrontal circuits, and how they, especially theta sequences, are linked to behavioral-timescale mechanisms for decision-making is unknown. To address these questions, we examined neuronal ensemble activity simultaneously in the hippocampus and PFC of rats during learning of a spatial working-memory decision task.

## Results

### Choice-specific neuronal sequences in CA1 and PFC during navigation decisions

We trained nine rats to learn a spatial working-memory task (*Figure 1A*). In this delayed alternation task, animals had to traverse a spatial delay section (i.e. common 'center stem'; no enforced delay) of a W-maze on each trial, and the critical memory demand of this task is to distinguish left (L) versus right (R) choices (*Figure 1A*): when the animals return inward from the side arm to the center reward well, they are required to remember their past choice between two possible locations (L vs. R arm; inbound reference-memory trial, *Figure 1A*, *left*); and have to choose the opposite side arm correctly for reward after running outward through the stem when facing the two upcoming options (outbound working-memory trial, *Figure 1A*, *right*) (*Jadhav et al., 2012*; *Kim and Frank, 2009*). This task is known to require both the hippocampus and PFC for learning (*Fernández-Ruiz et al., 2019*; *Jadhav et al., 2012*; *Kim and Frank, 2009*; *Maharjan et al., 2018*), and involves memory-guided decision-making (*Jadhav et al., 2016*; *Shin et al., 2019*; *Yu and Frank, 2015*). All subjects learned the task rules over eight training sessions (or epochs; 15–20 mins per sessions) in a single day, and performed with high levels of accuracy at the end of the training (*Figure 1B*; final performance: 92.5 ± 1.8% for inbound, 80.8 ± 2.8% for outbound, in mean ± SEM).

We used continuous and simultaneous recordings from ensembles of dorsal CA1 hippocampal and PFC neurons as rats learned this task (*Figure 1C* and *Figure 1—figure supplement 1*; mean ± SEM = 43.9 ± 7.6 CA1 place cells, 29.8 ± 5.6 PFC cells per session). As a result of spatially specific firing, sequences of CA1 and PFC cells successively activated on the timescale of seconds as the rat ran through a trajectory (i.e. behavioral sequences), as previously reported by several groups (*Frank et al., 2000*; *Fujisawa et al., 2008*; *Ito et al., 2015*; *Kinsky et al., 2020*; *Shin et al., 2019*; *Stout and Griffin, 2020*; *Wood et al., 2000*). Further, these sequences were reactivated on the timescale of hundreds of milliseconds during SWRs at the reward well (i.e., replay sequences; *Figure 1C*), confirming our previous findings (*Shin et al., 2019*). Notably, within each cycle of theta oscillations during navigation, which corresponds to expression of fast CA1 theta sequences (*Dragoi and Buzsáki, 2006*; *Foster and Wilson, 2007*; *Gupta et al., 2012*; *Skaggs et al., 1996*), we found that PFC cells were organized into sequences as well and could occur concurrently with the hippocampal sequences (*Figure 1C*). Given these observations, we further quantified these sequences and investigated their roles in decision making.

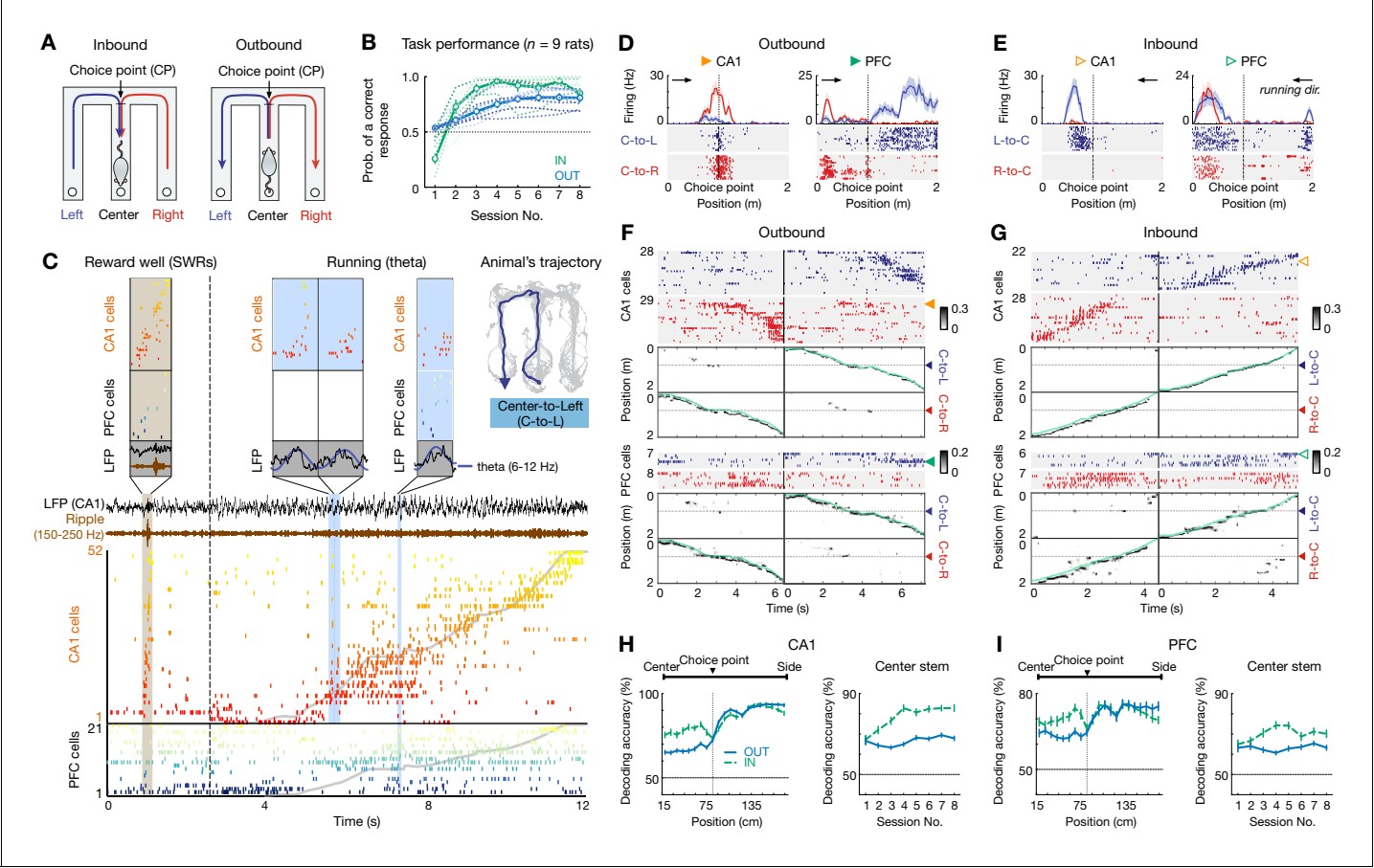

**Figure 1.** Choice-specific sequences in CA1 and PFC during memory-guided navigation decisions. (A) Diagrams of two possible past (inbound; *left*) and future (outbound; *right*) scenarios during a W-maze spatial alternation task. In this task, rats have to remember their past choice between two possible locations (Left vs. Right arms; *left*), and then choose the opposite arm correctly (*right*; see Materials and methods). CP: choice point. (B) Behavioral performance of all nine rats that learned the inbound (IN; green) and outbound (OUT; blue) components of the task over eight sessions within a single day. Dashed lines: individual animals. Thick lines with error bars: means with SEMs. Horizontal dashed lines: chance-level of 0.5. Prob.: probability. (C) Simultaneously recorded ensembles of CA1 and PFC cells, forming slow behavioral sequences spanning the entire trajectory (~8 s), fast replay sequences (~260 ms; brown shading), and fast theta sequences (~100–200 ms; blue shadings). Each row represents a cell ordered and color-coded by spatial-map center on the Center-to-Left (C–to–L) trajectory shown *top right*. Gray lines: actual position. Dashed vertical line: reward well exit. Black, brown, and dark blue lines: broadband, ripple-band, and theta-band filtered LFPs from one CA1 tetrode, respectively. (D–I) Choice-predictive representations of behavioral sequences in CA1 and PFC. (D and E) Four trajectory-selective example cells during (D) outbound and (E) inbound navigation (shadings: SEMs; black arrowheaded line indicates animal's running direction). (F and G) Example choice-specific behavioral sequences. For each plot pair, the *top* illustrates a raster of trajectory-selective cell assemblies ordered by spatial-map centers on the preferred trajectory; the *bottom* shows population decoding of animal's choice and locations at the behavioral timescale (bin = 120 ms; note that summed probability of each column across two trajectory types is 1). Green and yellow arrowheads indicate the example cells shown in (D) and (E). Color bar: posterior probability. Green lines: actual position. Blue and red arrowheads: the CP. Note that the rasters only show trajectory-selective cells, whereas the population decoding was performed using all cells recorded in a given region. (H and I) Behavioral sequences in (H) CA1 and (I) PFC predicted current choices. *Left*: decoding accuracy of current choice over locations (n = 9 rats×8 sessions); *Right*: decoding accuracy of current choice on the center stem across sessions. Note that the decoding performance is significantly better than chance (50%) over locations and sessions (all p's < 1e-4, rank-sum tests). Error bars: SEMs. OUT: outbound; IN: inbound.

The online version of this article includes the following source data and figure supplement(s) for figure 1:

**Source data 1.** Decoding accuracy at the behavioral timescale.

**Figure supplement 1.** Recording locations and behavioral-sequence representations of locations over learning.

First, we characterized how CA1 and PFC neurons encode choices on the behavioral timescale. We found that at the single-cell level, many CA1 and PFC cells exhibited strong preferential firing during navigation on L- versus R- side trajectory (trajectory-selective cells; mean ± SEM = 38.4 ± 1.2% in CA1, 23.5 ± 1.3% in PFC for inbound, 35.0 ± 0.7% in CA1, 20.9 ± 0.6% in PFC for outbound;

*Figure 1D and E*), consistent with prior reports (*Frank et al., 2000*; *Fujisawa et al., 2008*; *Ito et al., 2015*; *Kinsky et al., 2020*; *Shin et al., 2019*; *Stout and Griffin, 2020*; *Wood et al., 2000*). These trajectory-selective cells, when ordered by the peak firing on the preferred trajectory, form unique sequences for each choice type spanning the entire trial length at the behavioral timescale, including the common center stem prior to the choice point (CP) and the side arms after the CP (*Figure 1F and G*).

To directly assess the cell-assembly representation of the animals' choices, we used a memoryless Bayesian decoding algorithm (see Materials and methods; decoding bin = 120 ms) (*Shin et al., 2019*). We found that behavioral sequences in CA1 and PFC consistently predicted the animals' current location and choice well above chance level across all positions of a trial (*Fujisawa et al., 2008*; *Ito et al., 2015*; *Kinsky et al., 2020*; *Shin et al., 2019*; *Stout and Griffin, 2020*; *Wood et al., 2000*), even on the common center stem (*Figure 1H and I* and *Figure 1—figure supplement 1*). Notably, the decoding accuracy on the center stem significantly increased in CA1 (p=0.0003 and <1e-4 for outbound and inbound, respectively), and was stable in PFC across sessions (p's > 0.05 for outbound and inbound, Kruskal-Wallis test with Dunn's post hoc; *Figure 1H and I*, *right*), in agreement with our previous findings of an increased proportion of CA1 'splitter cells' (center-stem trajectory-selective cells) and a stable proportion of PFC 'splitter cells' over learning (*Shin et al., 2019*). In addition, the larger proportions of inbound splitter cells (*Shin et al., 2019*) also contributed to better decoding for inbound choices compared to that for outbound (*Figure 1H and I*). Nonetheless, behavioral sequences of CA1 and PFC cells represented unique choice types throughout the course of learning.

Therefore, these results suggest that choice information progresses through heterogeneous neuronal sequences in CA1 and PFC at the behavioral timescale as rats run along each trajectory. Importantly, these choice representations also provide distinguishable templates for the Bayesian decoder to identify and determine the content of fast theta sequences.

## Fast theta sequences in CA1 and PFC

Theta oscillations are prominent in CA1 during navigation, and CA1 cell assemblies are organized into theta sequences within single oscillation cycles. While it is not known whether ordered sequences of PFC cells occur during hippocampal theta oscillations, previous studies have shown that PFC cells phase-lock and phase-precess (i.e. spikes of a cell occur at progressively earlier theta phases as an animal move through its spatial field) to hippocampal theta oscillations (*Jadhav et al., 2016*; *Jones and Wilson, 2005a*; *Siapas et al., 2005*; *Sigurdsson et al., 2010*); further, theta-frequency synchrony and coherent spatial coding between the hippocampus and PFC is prominent during memory-guided navigation (*Benchenane et al., 2010*; *Gordon, 2011*; *Hasz and Redish, 2020*; *Jones and Wilson, 2005b*; *Sigurdsson et al., 2010*; *Zielinski et al., 2019*).

In order to statistically identify theta sequences in CA1 and PFC, the firing pattern within each candidate hippocampal theta cycle (≥5 cells active in a given brain region) during active running were analyzed using the Bayesian decoding approach (decoding bin = 20 ms), and the sequential structure of the Bayesian reconstructed positions was evaluated by shuffling procedures (see Materials and methods). Using this method, clear theta sequences were found in CA1 during inbound and outbound navigation across all sessions (*Figure 2A–C*), Intriguingly, significant theta sequences were also detected in PFC (*Figure 2D–F*). The prevalence of theta sequences in PFC was similar to that in CA1 (*Figure 2G*), although as expected, higher trajectory scores (suggesting more reliable timing of sequences) were observed in CA1 compared to PFC (*Figure 2H*). Furthermore, the trajectory scores and slopes (sequence speed) in both regions increased over learning (*Figure 2H and I*). Finally, consistent with previous studies in CA1 (*Gupta et al., 2012*; *Wikenheiser and Redish, 2015*; *Zheng et al., 2016*), the majority of CA1 (~70%) and PFC (~60%) theta sequences successively represented past, present, and future locations within each theta cycle (i.e. forward sequences; *Figure 2J* and *Figure 2—figure supplement 1A–C*), at a velocity approximately 4–15 times faster than an animal's true running speed (*Figure 2I*). These properties of CA1 and PFC theta sequences were replicated with a 10 cell threshold and different shuffling procedures (*Figure 2—figure supplement 1*), suggesting that the observation of theta sequences in CA1 and PFC was not a trivial result of our decoding methodology.

To test whether theta phase precession can account for the occurrence of theta sequences, we performed a phase-jitter analysis (*Foster and Wilson, 2007*), in which we randomly chose a phase of

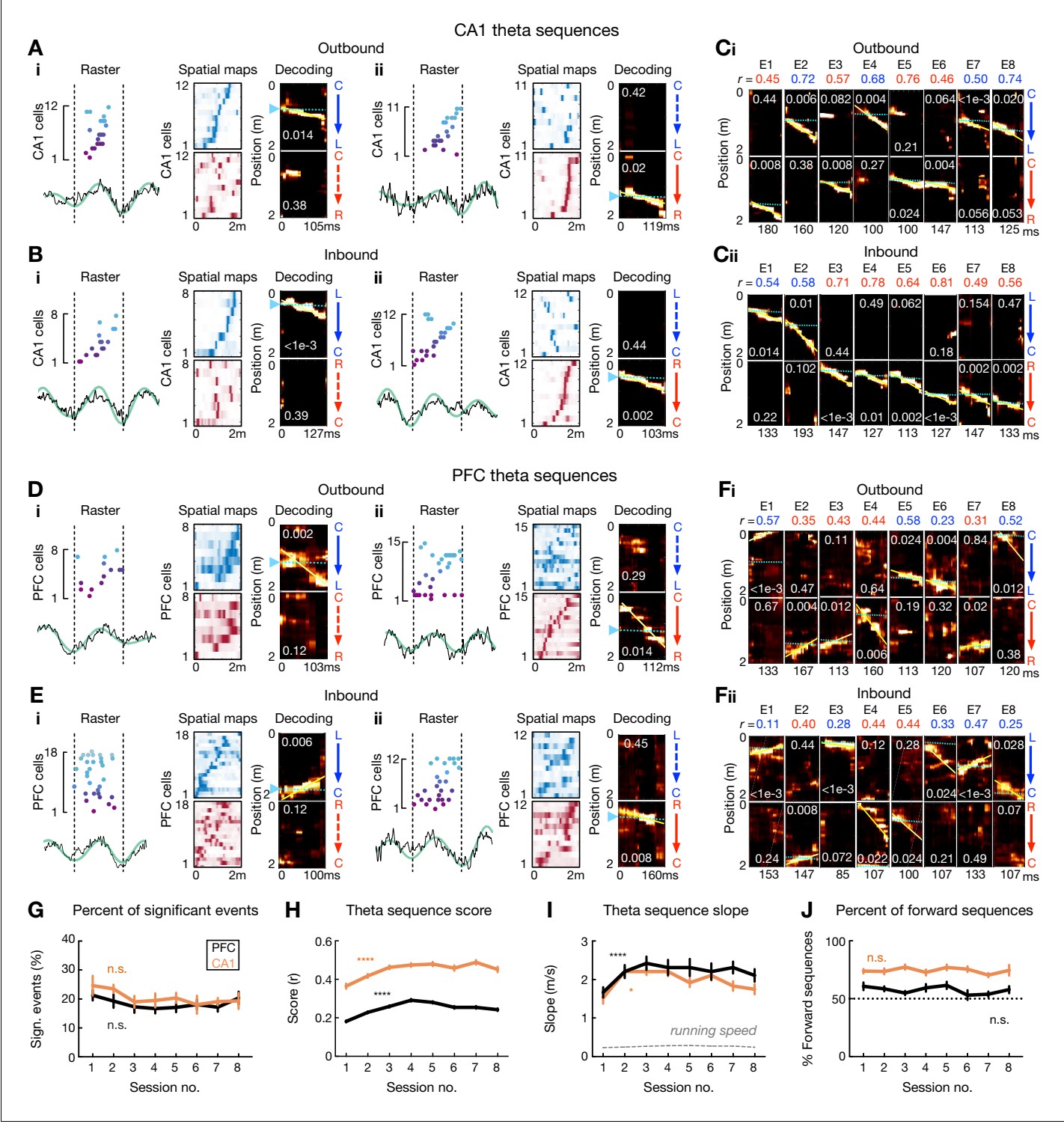

**Figure 2.** Theta sequences in CA1 and PFC over learning. (**A and B**) Four examples of theta sequences in CA1 for (**A**) outbound and (**B**) inbound trajectories. *Left*: spikes ordered and color coded by spatial-map center on the decoded trajectory (see *Right*) over a theta cycle. Broadband (black) and theta-band filtered (green) LFPs from CA1 reference tetrode shown below. *Middle*: corresponding linearized spatial firing rate maps (blue colormap for L-side trajectory, red colormap for R-side trajectory). *Right*: Bayesian decoding with *p*-values based on shuffled data denoted (see Materials and methods). Yellow lines: the best linear fit on the decoded trajectory. Cyan lines and arrowheads: actual position. Note that summed probability of each column across two trajectory types is 1. (**C**) Example CA1 theta sequences across eight sessions (or epochs, E1 to E8). Each column of plots represents a theta-sequence event. Sequence score (*r*) on the decoded trajectory denoted. (**D and E**) Four examples of theta sequences in PFC

*Figure 2 continued on next page*

*Figure 2 continued*

for (D) outbound and (E) inbound trajectories. Data are presented as in (A) and (B). LFPs are from CA1 reference tetrode. (F) Example PFC theta sequences across eight sessions. Data are presented as in (C). (G) Percent of significant theta sequences out of all candidate sequences didn't change significantly over sessions (Session 1 vs. 8: p's > 0.99 for CA1 and PFC, Friedman tests with Dunn's post hoc). Data are presented as mean and SEM. Black line: PFC; Orange line: CA1. (H) Theta sequence scores (*r*) improved over sessions (****p's < 1e-4 for CA1 and PFC, Kruskal-Wallis tests with Dunn's post hoc). Data are presented as median and SEM. (I) Theta sequence slopes increased over sessions (*p=0.0427 for CA1, and ****p<1e-4 for PFC, Kruskal-Wallis tests with Dunn's post hoc). Dashed gray lines: mean ± SEMs of animals' running speed (small error bars may not be discernable). Data are presented as median and SEM. (J) Percent of forward theta sequences didn't change significantly over sessions (p=0.56 and 0.41 for CA1 and PFC, Friedman tests). Data are presented as mean and SEM.

The online version of this article includes the following source data and figure supplement(s) for figure 2:

**Source data 1.** Percent of significant theta sequences, sequence score, sequence slope, and percent of forward theta sequences.
**Figure supplement 1.** Detection of theta sequences in CA1 and PFC is statistically robust.
**Figure supplement 2.** Theta phase precession does not account for the full extent of observed theta sequences in CA1 and PFC.

a spike from the distribution of possible phases of that cell in the position bin and shift the spike time accordingly for each candidate event. We found that sequence scores of actual events were significantly greater than those of shuffles in both CA1 and PFC (*Figure 2—figure supplement 2*), and thus theta phase precession does not account for the full extent of observed theta sequences in CA1 and PFC.

## Look-ahead of theta sequences during outbound versus inbound navigation

Prior work has shown that hippocampal forward theta sequences encode paths ahead of the animal, potentially providing a 'look-ahead' prediction of upcoming locations (*Dragoi and Buzsáki, 2006*; *Foster and Wilson, 2007*; *Gupta et al., 2012*; *Lisman and Redish, 2009*; *Maurer and McNaughton, 2007*; *Skaggs et al., 1996*; *Wikenheiser and Redish, 2015*). To investigate how these theta sequences related to animals' memory state and upcoming behavior, we examined the Bayesian reconstructed positions of forward-shifted candidate theta sequences in different theta-phase bins during reference-memory-guided inbound versus working-memory-guided outbound navigation. We found that the positions behind the actual location of the animal were decoded with higher probability during inbound than outbound navigation in both CA1 and PFC, (*Figure 3A–D*), implying that inbound sequences started farther behind and outbound sequences sweep father ahead of the current animal position. To further confirm this result, we directly compared the start and end positions of each significant theta sequence, and indeed, forward sequences in both CA1 and PFC started farther behind the actual position of the animal during inbound navigation, whereas they ended farther ahead of the animal during outbound navigation (*Figure 3E and F*). A possibility that could bias this look-ahead difference is that the choice point may represent a salient behavioral state, acting as an attractor state 'pulling' representations toward it. This hypothesis would predict a stronger effect when an animal approaches the choice point compared to the reward well. However, we obtained similar results in these two situations (*Figure 3—figure supplement 1*), indicating the difference in theta-sequence look-ahead is not a simple result of the attractor behavioral states of the choice point or reward well.

We then investigated potential neuronal mechanisms underlying this shift in ahead-sequence length. On a single-cell level, previous theoretical and experimental evidence has suggested that the initial tail of asymmetric spatial fields allows cells with fields ahead of an animal's position to fire during earlier theta cycles, which results in 'look-ahead' of theta sequences (*Burgess and O'Keefe, 2011*; *Mehta et al., 2002*; *Mehta et al., 2000*; *Skaggs et al., 1996*; *Wikenheiser and Redish, 2015*). We therefore examined the relationship between firing field asymmetry and the shift in ahead-sequence length during outbound and inbound navigation (*Figure 4A–D*). We found that while the asymmetry developed with experience, as reported previously in CA1 (*Figure 4A–C*; *Mehta et al., 2000*), working-memory-guided outbound navigation was associated with fields with a more extended initial tail compared to inbound travel for both CA1 and PFC (*Figure 4A–C*). This effect was also stronger for trajectory-selective than non-selective cells in both regions (*Figure 4D*).

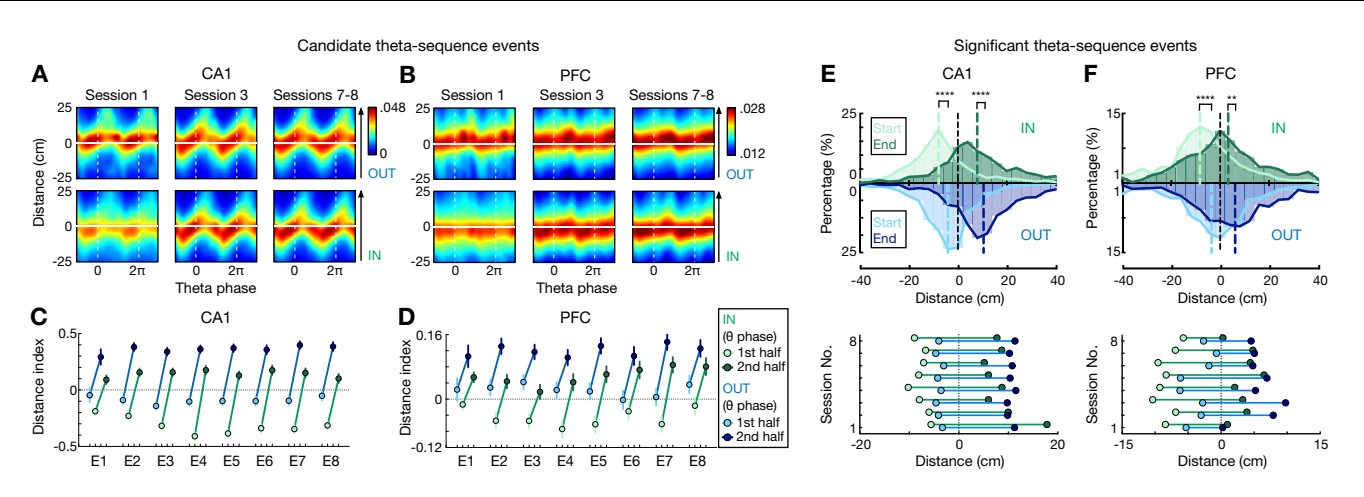

**Figure 3.** Look-ahead of CA1 and PFC theta sequences differs during outbound versus inbound navigation. (**A and B**) Theta sequences representing past, current, and future locations on outbound (OUT) and inbound (IN) trajectories in (**A**) CA1 and (**B**) PFC. Each plot shows the averaged Bayesian reconstruction of all forward candidate theta sequences (sequence score $r > 0$), replicated over two theta cycles for visualization, relative to current position ($y = 0$; $y > 0$: ahead, or future location; $y < 0$: behind, or past location). Color bar: posterior probability. Arrowheaded line: animal's running direction. (**C and D**) Distance index in (**C**) CA1 and (**D**) PFC across eight sessions (E1–E8), compared the posterior probabilities on future versus past locations ($<0$, biased to past; $>0$, biased to future). 1st half of theta phases (light circles): $-\pi$ to 0; 2nd half of theta phases (dark circles): 0 to $\pi$. Error bars: 95% CIs. (**E and F**) Distributions for start and end of reconstructed trajectories relative to actual position ($x = 0$) of all significant forward theta sequences in (**E**) CA1 and (**F**) PFC. *Top*: Histograms show the distributions across all eight sessions (dashed vertical lines: median values). *Bottom*: Averaged trajectory start (light circles) and end (dark circles) positions in individual sessions. ****$p < 1e-4$, **$p = 0.007$, Kolmogorov-Smirnov test.

The online version of this article includes the following source data and figure supplement(s) for figure 3:

**Source data 1.** Start and end of reconstructed trajectories of all significant theta sequences.

**Figure supplement 1.** Look-ahead of theta sequences was similar during approach to choice point and approach to reward well.

Further, a prominent model suggests that phase precession of individual neurons may give rise to the look-ahead forward sweep of theta sequences in CA1 (*Dragoi and Buzsáki, 2006*; *Skaggs et al., 1996*). We therefore examined if the shift in ahead-sequence length was a consequence of change in phase precession speed during inbound versus outbound navigation. Consistent with previous studies (*Jones and Wilson, 2005a*; *Skaggs et al., 1996*), we found CA1 and PFC cells phase-precessing to hippocampal theta oscillations (*Figure 4E and F*). However, similar slopes of theta phase precession were observed for inbound and outbound navigation in both regions (*Figure 4G and H*), which therefore cannot simply explain the difference in inbound and outbound look-ahead of theta sequences.

Together, these results suggest that beyond the pure sensory features of the environment, memory demands influenced the look-ahead properties of theta sequences in both CA1 and PFC, and the increased look-ahead distance during working-memory-guided outbound navigation allows the animal to represent future locations earlier in the trajectory, which can aid in decision-making.

## Theta sequences support vicarious memory recall

How do CA1 and PFC theta sequences relate to the animals' upcoming choices when multiple options are available, and do they encode current goal throughout navigation similar to the behavioral sequences (*Figure 1*)? Prior work has reported that hippocampal population activity at the theta timescale can represent alternatives, which potentially supports deliberation (*Johnson and Redish, 2007*; *Kay et al., 2020*), and such a population code is linked to single-cell cycle skipping, in which cells fire on alternate theta cycles (*Kay et al., 2020*). Therefore, we first examined single-cell firing at the theta timescale in CA1 and PFC. Consistent with previous studies (*Dragoi and Buzsáki, 2006*; *Kay et al., 2020*), we observed normal theta rhythmic (i.e. non-skipping; firing on adjacent cycles) and cycle skipping (i.e. firing on every other cycle) cells in CA1 (*Figure 5A and B*). Intriguingly, we found that a large proportion of theta-modulated cells in PFC also fired in regular

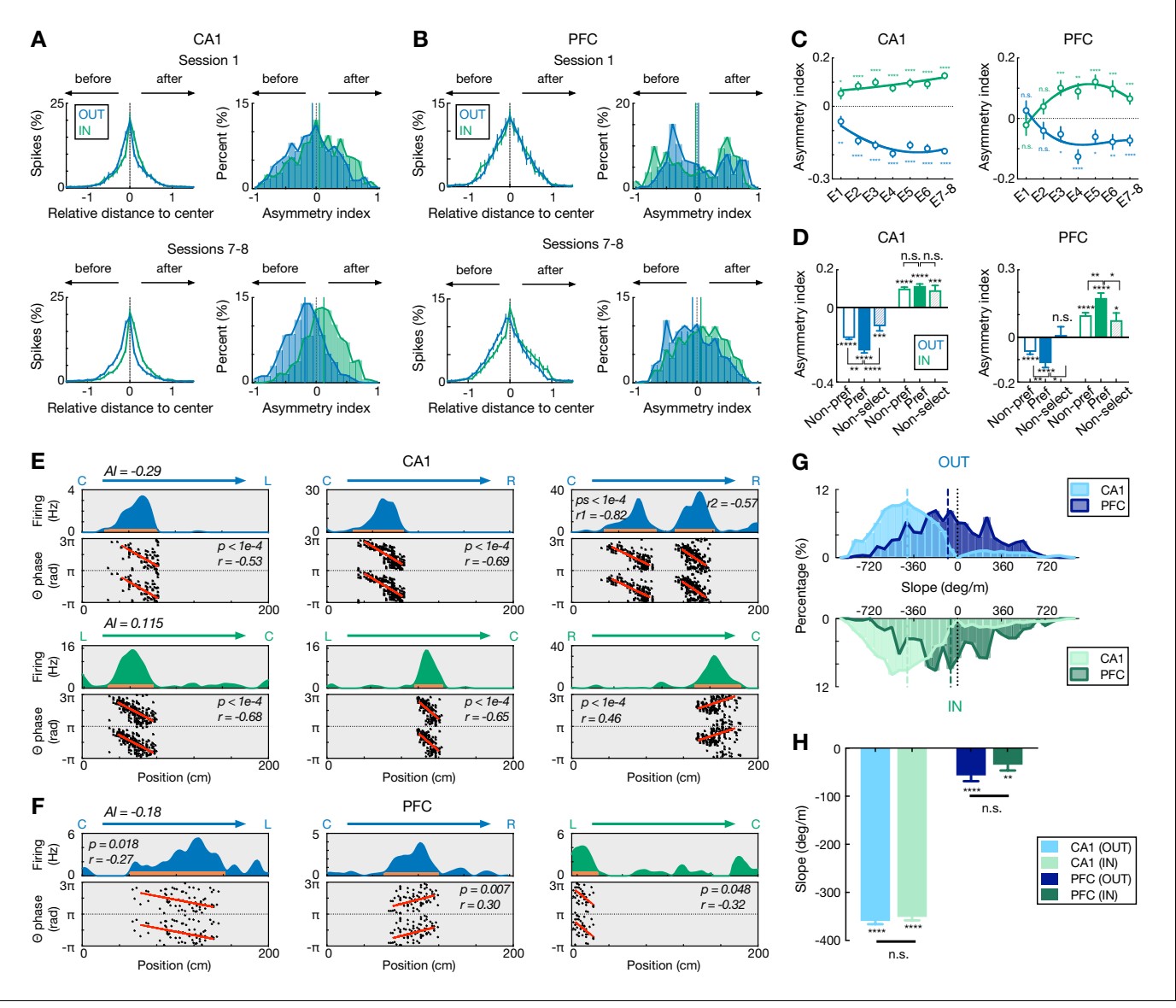

**Figure 4.** Comparisons of spatial-field asymmetry and theta phase precession during outbound versus inbound navigation in CA1 and PFC. (A and B) Spatial-field asymmetry during Session 1 (*top*) and Sessions 7–8 (*bottom*) in (A) CA1 and (B) PFC. Blue: outbound fields (OUT); Green: inbound fields (IN). *Left*: Averaged firing rate relative to field center (*x* = 0) across all cells in the given sessions (error bars: SEMs). *Right*: Distributions of spatial-field asymmetry index (colored vertical lines: mean values). See also single-field examples in (E) and (F). (C) Spatial-field asymmetry index across sessions (****p<1e-4, ***p<0.001, **p<0.01, *p<0.05, n.s. p>0.05, signed-rank tests compared to 0). Lines are derived from polynomial fits. (D) Trajectory-selective cells exhibit highly asymmetric fields on the preferred (Pref) trajectory compared to the non-preferred trajectory (Non-pref). Non-select: non-selective cells. *P*-values for each condition derived from signed-rank tests compared to 0; *p*-values across conditions derived from rank-sum tests (****p<1e-4, ***p<0.001, **p<0.01, *p<0.05, n.s. p>0.05). (E and F) Single-cell examples of theta phase precession in (E) CA1 and (F) PFC for outbound (blue) and inbound (green) trajectories. For each example, linearized firing fields are shown on the *top* (trajectory type denoted above); spike theta phases against positions (i.e. phase precession) within individual spatial fields (indicated by an orange bar on the firing-field plot) are shown on the *bottom* (phases are plotted twice for better visibility; red lines represent linear-circular regression lines; linear-circular correlation coefficient *r* and its p-value denoted). AI: spatial-field asymmetry index. (G) Distributions of phase precession slopes for outbound and inbound fields. *Top*: outbound; *bottom*: inbound. Vertical lines: median values. (H) Phase precession slopes were similar during outbound versus inbound navigation in CA1 and PFC (n.s., p's > 0.99, Kruskal-Wallis test with Dunn's post hoc), and biased toward negative values (****p<1e-4, **p=0.0012, signed-rank tests compared to 0). This bias is stronger in CA1 than PFC (p's < 1e-4, Kruskal-Wallis test with Dunn's post hoc). Only fields with significant phase precession (see Materials and methods) are shown (mean ± SEM = 46.0 ± 2.8% in CA1, 11.2 ± 1.7% in PFC for inbound, and 48.8 ± 3.3% in CA1, 10.3 ± 1.2% in PFC for outbound). Error bars: SEMs.

The online version of this article includes the following source data for figure 4:

**Source data 1.** Asymmetry index and phase-precession slope.

alternation during hippocampal theta oscillations (i.e. cycle skipping; Inbound: 53.4%, 95 out of 178; outbound: 49.7%, 91 out of 183; *Figure 5C and D*).

Given the single-cell property of cycle-skipping identified above in both regions, we next examined how populations of CA1 and PFC cells encoded choices during theta sequences. Indeed, we found that CA1 theta sequences can encode alternatives (*Figure 6A*), as previously reported (*Johnson and Redish, 2007*; *Kay et al., 2020*). However, when we examined the representations of choices by theta sequences along each trajectory, we found these representations were distinct in CA1 and PFC during decision-making periods prior to the choice point (CP). Before the CP (corresponding to periods on the center stem for outbound navigation; *Figure 6B*), CA1 theta sequences serially encoded both alternative trajectories (*Figure 6C and G* and *Figure 6—figure supplements*

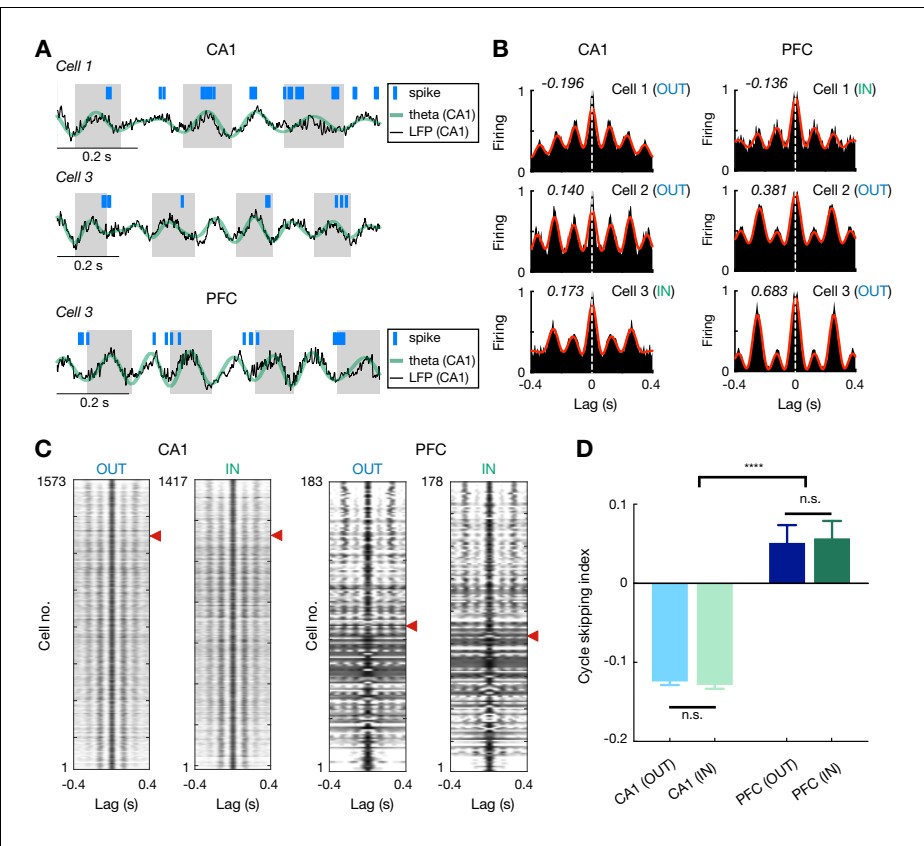

**Figure 5.** Theta cycle skipping in CA1 and PFC. (**A**) Spike rasters of example cells during CA1 theta oscillations. *Top*: a non-skipping (firing on adjacent cycles) CA1 cell; *Middle*: a cycle-skipping (firing on every other cycles) CA1 cell; *Bottom*: a cycle-skipping PFC cell. Black and green lines: broadband and theta-filtered LFPs from CA1 reference tetrode, respectively. (**B**) Auto-correlograms (ACGs) of three example single cells in CA1 (*left*) and PFC (*right*). Each plot is of data from a single type of maze travel (outbound or inbound; travel type denoted). For each plot, cycle skipping index (CSI) is denoted on the upper left corner (CSI < 0: firing on adjacent cycles; CSI > 0: cycle skipping), and cell number with maze travel type (IN or OUT) matched to (**A**) is denoted on the upper right corner. Red line: low-passed (1–10 Hz) ACG to measure CSI (see Materials and methods). Note that cells on the bottom two rows exhibit cycle skipping, with CSI > 0. (**C**) ACGs of all theta-modulated cells in CA1 (*left*) and PFC (*right*) ordered by their CSIs (high to low from top to bottom). Red arrowheads indicate division between cells with CSI > 0 (above) vs. <0 (below). Each row represents a single cell with one type of maze pass (outbound or inbound). Only cells with theta-modulated ACGs are shown (see Materials and methods). Note that the proportion of theta-cycle skipping cells in CA1 is consistent with that reported in previous studies (*Kay et al., 2020*). (**D**) CSI of theta-modulated cells didn't differ significantly on outbound versus inbound trajectories for each region (n.s., p's > 0.99 for CA1 and PFC), but was larger in PFC than CA1 (****p<1e-4, Kruskal-Wallis tests with Dunn's post hoc). Data are presented as mean and SEMs.

The online version of this article includes the following source data for figure 5:

**Source data 1.** Cycle skipping index.

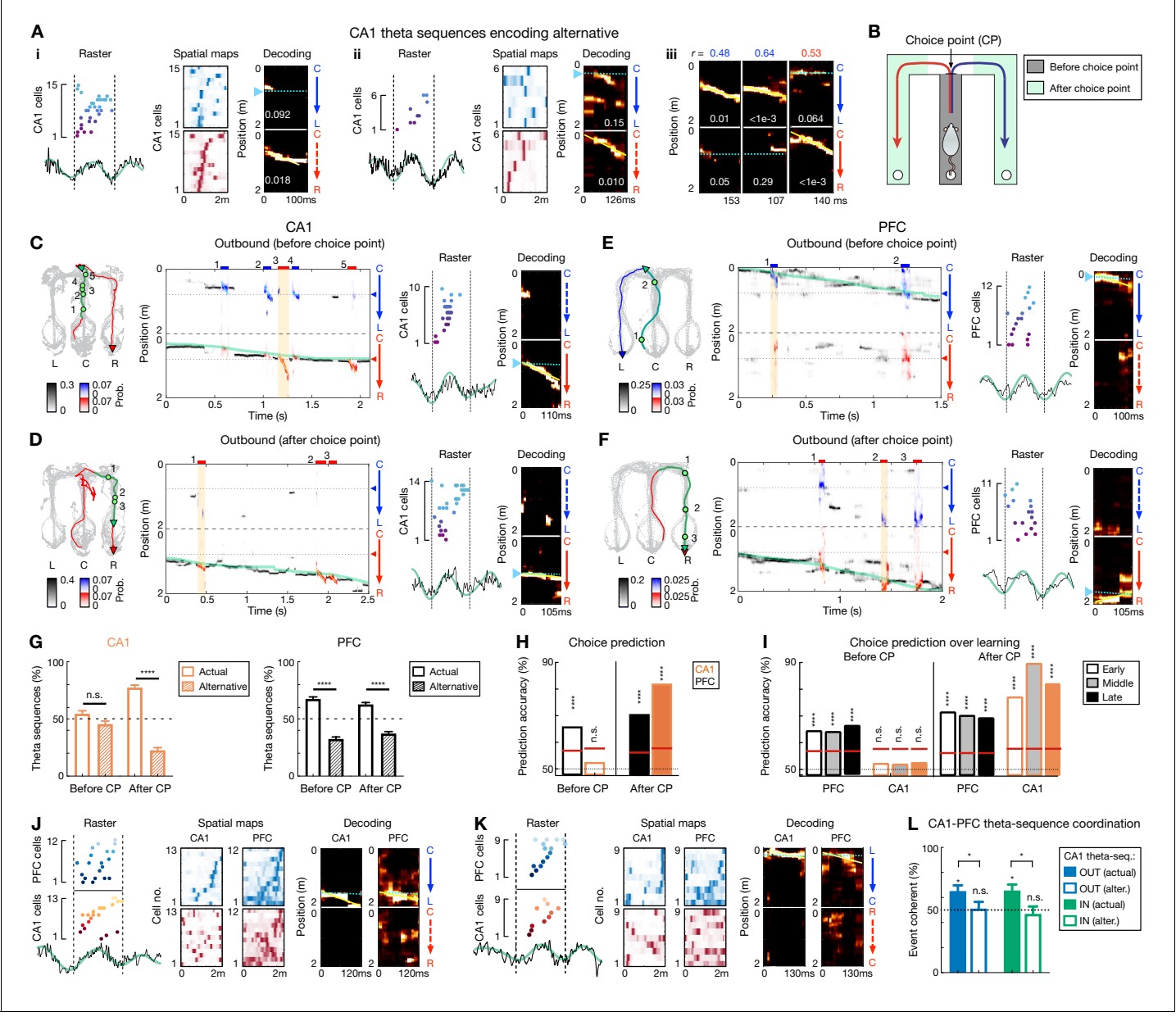

**Figure 6.** Theta-sequence representations of behavioral choices in CA1 and PFC. (**A**) Single-event examples of CA1 theta sequences encoding alternative. Two detailed example sequences are shown in (Ai) and (Aii) (presented as in *Figure 2A*). Three more examples with decoding plots only are shown in (Aiii) (presented as in *Figure 2C*). (**B**) Diagram showing two task segments of a behavioral trial (or trajectory): before choice point (gray shading), and after choice point (green shadings). (**C–F**) Four decoding examples for before and after choice point during outbound navigation in (**C** and **D**) CA1 and (**E** and **F**) PFC. *Left*: Animal's behavior. Green line: the trajectory pass shown on the *middle*; Blue/red arrowheaded line: currently taken trajectory. Green circles: locations where theta-sequence events occurred (numbered corresponding to *middle*). *Middle*: Decoding plots. Data are presented as in *Figure 1F and G* (bin = 120 ms), except that whenever a theta sequence was detected, the decoding was performed on the theta timescale (bin = 30 ms) and color-coded by trajectory type for clarity (red or blue: R-side or L-side trajectory; bars above show decoded identity and timing of each event). Note that at both timescales, summed probability of each column across two trajectory types is 1. Yellow shading: example event with detailed view shown on the *right*. Prob.: probability. (**G**) Percent of theta sequences representing actual or alternative choice before and after choice point (CP) in CA1 (*left*) and PFC (*right*) (****p<1e-4, n.s. p>0.05, session-by-session rank-sum paired tests). Error bars: SEMs. (**H**) Trial-by-trial theta-sequence prediction of choice (****p<0.0001, n.s., p>0.05, trial-label permutation tests). Red horizontal lines: chance levels (i.e. 95% CIs of shuffled data) calculated by permutation tests. CP: choice point. (**I**) Theta-sequence prediction persists over sessions. Early: Sessions 1–3; Middle: Sessions 4–5; Late: Sessions 6–8. Data are presented as in (**H**). Only correct trials are shown in (**H**) and (**I**). CP: choice point. (**J–L**) Coherent CA1-PFC theta sequences biased to actual choice. (**J** and **K**) Two examples of coherent CA1-PFC theta sequences. (**L**) Percent of coordinated CA1-PFC theta sequences coherently representing actual vs. alternative choices (for each condition from left to right: p=0.0312, 0.94, 0.0312, 0.48, signed-rank test

*Figure 6 continued on next page*

*Figure 6 continued*

compared to 50%; for comparisons between two conditions from left to right, p=0.0312 and 0.0469, rank-sum tests). OUT: outbound; IN: inbound. Alter.: alternative choice. Error bars: SEMs.

The online version of this article includes the following source data and figure supplement(s) for figure 6:

**Source data 1.** Choice representation of theta sequences, and number of coordinated theta sequences for each session.
**Figure supplement 1.** Additional examples of theta-sequence representations of behavioral choices.
**Figure supplement 2.** Theta-sequence coding for behavioral choices during inbound navigation and additional controls.

*1–2*; *Video 1*). In contrast, PFC theta sequences preferentially encoded the animal's current choice (*Figure 6E and G* and *Figure 6—figure supplements 1–2*). After the choice was made (i.e., after CP; *Figure 6B*), as expected, both CA1 and PFC theta sequences preferentially encoded the animal's current choice as the animal's ran down the track toward reward (*Figure 6D, F and G* and *Figure 6—figure supplements 1–2*). Note that after CP, CA1 theta sequences representing the alternative choice only constituted a minority of the total sequences, reminiscent of the previous findings of a small proportion of hippocampal theta sequences representing the alternative running direction on a linear track (*Feng et al., 2015*; *Wang et al., 2020*), and encoding locations far away from an animal's current position (*Gupta et al., 2012*; *Wikenheiser and Redish, 2015*).

The choice representations of theta sequences were robust across sessions (*Figure 6G*), enabling trial-by-trial prediction of decisions, in which upcoming choice was decoded by PFC theta sequences well above chance before the CP, whereas CA1 theta sequences encoded actual and alternative available paths equivalently before the CP (*Figure 6H and I*). Similar results were found for inbound trials (*Figure 6—figure supplements 1–2*; *Video 2*). Note that these results cannot be accounted for by similar spatial-map templates for L versus R choices on the center stem, because spatial-map activity can decode L versus R choices well above chance within the center stem in both CA1 and PFC, and this decoding accuracy was in fact higher for CA1 than PFC (*Figure 1H and I*). Furthermore, similar effects were found after controlling for different shuffling procedures (*Figure 6—figure supplement 2A*) and examining the last theta sequence on the center stem (*Figure 6—figure supplement 2D*).

Next, we asked if PFC theta sequences encoded choices that were coherent with CA1 sequences within single theta cycles. We detected PFC theta sequences simultaneously with CA1 theta sequences for a subset of theta

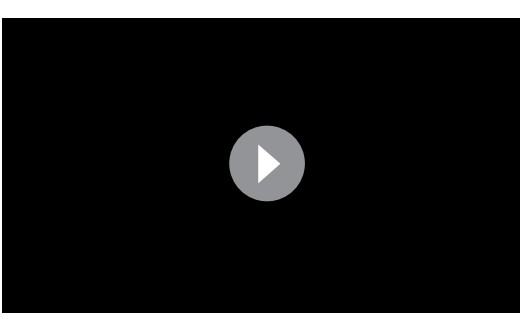

**Video 1.** Slow-motion video of behavioral-timescale and theta-sequence decoding in CA1 when the rat is running a Center-to-Right outbound trajectory. The video plays 7.5 times slower than real time. To better visualize fast theta sequences, when a significant theta sequence is detected, the video plays 15 times slower. Audio represents spiking of all example units shown on the raster (*top left*; each spike was correspondingly sped up 7.5 times around spike detection for better perception). *Bottom left*: Decoding plot. For each theta sequence, reconstruction of only the decoded trajectory was shown for clarity. *Right*: Behavioral video. Green circle: true position. White circle with a pair of arrowheads: estimated position decoded at the behavioral timescale (arrowhead colors indicate trajectory type, blue for L-side trajectory, red for R-side trajectory; solid arrowheads: decoded trajectory type. hollow ones: the alternative). Note that the raster only shows cells that participated in theta sequences for better visualization, whereas the decoding was performed using all place cells recorded.
https://elifesciences.org/articles/66227#video1

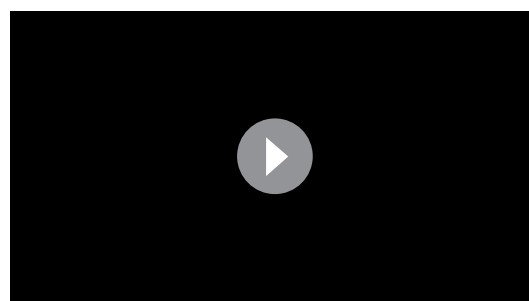

**Video 2.** Slow-motion video of behavioral-timescale and theta-sequence decoding in CA1 when the rat is running a Right-to-Center inbound trajectory. The video is presented as in *Video 1*.
https://elifesciences.org/articles/66227#video2

cycles (*Figure 6J and K*; mean ± SEM = 10.53 ± 1.42%, and 11.17 ± 1.52% out of significant sequences for outbound and inbound, respectively; p=0.78 comparing outbound vs. inbound proportions, rank-sum test). Among these synchronous sequence events, when CA1 theta sequences represented the actual choice, PFC sequences were also significantly biased to the actual choice, whereas this coherent CA1-PFC representation was not observed when CA1 theta sequences represented the alternative choice (*Figure 6L*). Note that this result cannot be simply accounted for by the fact that the majority of PFC theta sequences depict the actual choice, assuming independence of CA1 and PFC theta sequences (see Materials and methods).

Taken together, these results suggest that CA1-PFC theta sequences occurred in tandem with, but distinct from, behavioral sequences for choice representations, and that CA1-PFC theta sequences underlie a novel mechanism that supports vicarious memory recall on a fast timescale for deliberative decision-making.

### CA1 and PFC sequences during incorrect trials

While we found a clear relationship between CA1 and PFC sequences at both behavioral and theta timescales for upcoming decisions, it remained unclear if these contributed to correct versus erroneous decisions. We therefore analyzed neural activity during correct versus incorrect trials, with incorrect trials corresponding to erroneous outbound navigation to the same side arm as the past inbound visit (*Figure 7A*). We found that sequential firing that occurred at the behavioral timescale during incorrect trials was similar to that during correct trials (*Figure 7B*), and the decoding accuracy for the chosen side was comparable for correct and incorrect trials (*Figure 7C*; inbound trials for the incorrect condition were considered as the one right before an incorrect outbound trial). Furthermore, CA1 and PFC theta-sequence prediction of upcoming choice was also similar for correct and incorrect trials (*Figure 7D*).

Correct versus incorrect trials did not differ in running speed and theta power (*Figure 7—figure supplement 1A–B*). CA1-PFC theta coherence and the strength of single-cell phase-locking to theta oscillations during navigation, which have been proposed to support spatial working memory (*Benchenane et al., 2010*; *Gordon, 2011*; *Jones and Wilson, 2005b*; *Sigurdsson et al., 2010*), were also similar between correct versus incorrect trials (*Figure 7—figure supplement 1C–L*). Therefore, both theta-linked phenomena at the two timescales likely represented maintenance mechanisms for working memory and decision-making, making it plausible that incorrect destinations were chosen prior to embarking on trajectories from the center well.

We therefore examined replay sequences during SWRs in the inter-trial periods prior to trajectory onset (*Figure 1C*). Previously, we have reported that CA1 replay sequences, similar to its theta sequences reported here, underlie deliberation between actual and alternative choices, whereas CA1-PFC reactivation represents actual choice for current trials (*Shin et al., 2019*). Here, we confirmed these observations (*Figure 7E* and *Figure 7—figure supplement 2*; *Video 3*). Importantly, using CA1-PFC reactivation strength for actual versus alternative choices preceding the correct and incorrect trials (see Materials and methods), we could predict correct and incorrect responses significantly better than chance (*Figure 7F*), indicating impaired CA1-PFC reactivation prior to incorrect outbound navigation. These results suggest that CA1-PFC replay sequences during awake SWRs prime initial navigation decisions, which are further maintained by theta-sequence and trajectory-selective mechanisms during retention periods on a trial-by-trial basis, underlying successful performance in the working memory task.

## Discussion

In this work, we discovered theta sequences, theta cycle skipping and theta-sequence prediction of behavioral choices in PFC. These prefrontal phenomena follow a succession of results on hippocampal sequences, but our study points to different yet complementary roles of prefrontal and hippocampal sequences at multiple timescales. By dissecting fast cognitive-timescale sequences from slow behavioral-timescale sequences in a spatial working-memory and navigation task, these findings thus provide a unified framework that integrates hippocampal and prefrontal mechanisms of multi-timescale cell-assembly dynamics for memory-guided decision-making.

First, during delay periods of the spatial navigation task on the center stem, choice information was maintained by behavioral-timescale sequences in CA1 and PFC on correct as well as incorrect

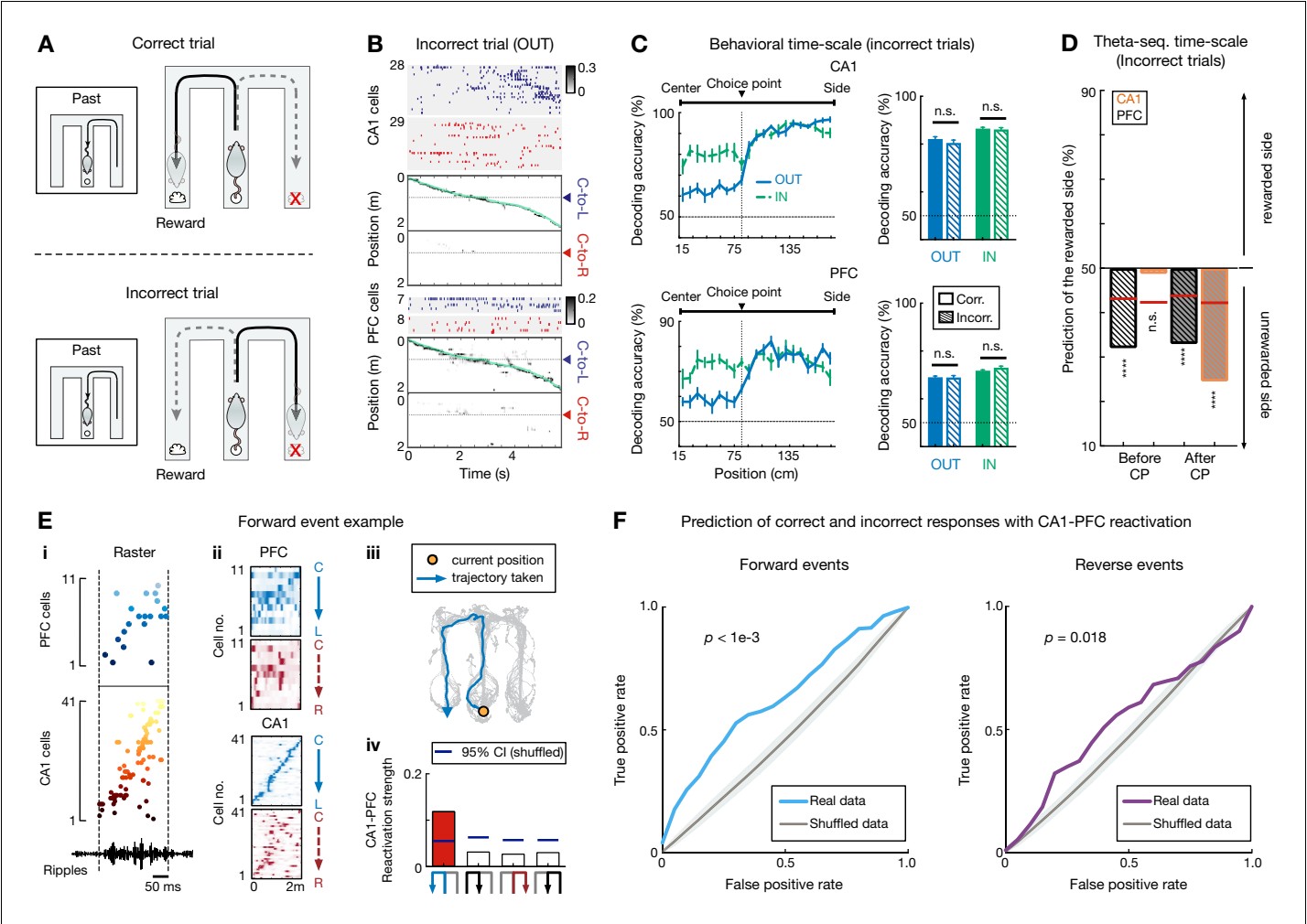

**Figure 7.** Choice representations of replay sequences, but not behavioral and theta sequences, were altered in CA1 and PFC during incorrect trials. (**A**) Illustration of a set of correct (*top*) and incorrect (*bottom*) trials. For incorrect trial, the actual choice is the unrewarded side. (**B and C**) Behavioral sequences encoded current choice during incorrect trials. (**B**) Rasters and population decoding during an incorrect outbound trial. Data are taken from the same session and animal and presented as in *Figure 1F*. (**C**) Choice decoding accuracy during incorrect trials is not significantly different from that during correct trials in CA1 (n.s., p>0.99 for outbound,=0.67 for inbound) and PFC (n.s., p>0.99 for outbound,=0.11 for inbound; Friedman tests with Dunn's post hoc). Corr.: correct trials; Incorr.: incorrect trials. Inbound trials for the incorrect condition were taken from the one right before an incorrect outbound trial (i.e. 'Past' trial of the diagram shown in A). (**D**) Choice representations of theta sequences were similar during incorrect and correct (see *Figure 6H*) trials. Black bars are for PFC theta sequences, orange bars are for CA1. CP: choice point. (**E**) Example forward CA1-PFC replay sequences representing actual future choice (see also *Figure 1C* for this event with example cells, and ripples from a different tetrode). (**E_i**) Ordered raster plot during a SWR event (black line: ripple-band filtered LFPs from one CA1 tetrode). (**E_ii**) Corresponding spatial firing rate maps. (**E_iii**) Actual (immediate future) trajectory (orange circle: current position when replay sequences occurred). (**E_iv**) Reactivation strength (trajectory schematics on the *bottom*). Blue horizontal lines: 95% CIs computed from shuffled data. Red bar: the decoded trajectory. See *Figure 7—figure supplement 2C* for an example of reverse CA1-PFC replay sequences. (**F**) CA1-PFC replay strength predicts correct and incorrect responses. *Left*: Prediction using replay strength of CA1-PFC forward events. *Right*: Prediction using replay strength of CA1-PFC reverse events. ROC curves were computed for the SVM classifiers (*p*-value from trial-label shuffling denoted; see Materials and methods). Shadings: SDs.

The online version of this article includes the following source data and figure supplement(s) for figure 7:

**Source data 1.** Decoding accuracy at the behavioral timescale for correct versus incorrect trials.
**Figure supplement 1.** Speed, theta power, coherence, and phase-locking during correct versus incorrect outbound trials.
**Figure supplement 2.** CA1-PFC reactivation strength during correct versus incorrect trials.

trials. These sequences are contextually modulated by current journeys, and can enable choice-related information processing on a behavioral timescale for planning actions (*Harvey et al., 2012*; *Ito et al., 2015*). In addition, we determined the existence of compressed-timescale theta sequences

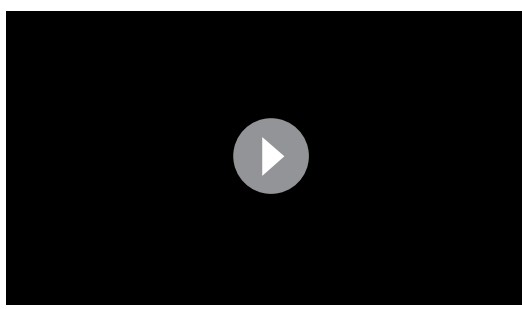

**Video 3.** Video of CA1 replay sequences representing possible future choices. The video is displayed in real time, but two times slower during immobility at the reward well to better visualize fast replay sequences. *Top left*: Raster of example CA1 cells ordered and color coded by place field center on the actual future trajectory (C-to-L). *Bottom left*: Raster of the same CA1 population shown on top left, but ordered and color coded by place field center on the alternative future trajectory (C-to-R). *Right*: Behavioral video. Green circle: true position. Large arrowhead: estimated location at the behavioral timescale. Due to the long immobility period at the reward well (9.58 s), only 1 s around each replay event detected was shown. When a replay sequence is detected, the decoded trajectory is represented by an arrowheaded line (colored according to the trajectory type, blue for L-side trajectory, red for R-side trajectory).
https://elifesciences.org/articles/66227#video3

in PFC, similar to previously described CA1 theta sequences (*Dragoi and Buzsáki, 2006*; *Foster and Wilson, 2007*; *Kay et al., 2020*; *Skaggs et al., 1996*). These transient theta sequences in CA1 and PFC were nested within behavioral-timescale sequences during navigation. However, in contrast to trajectory-dependent firing sequences at the behavioral-timescale, theta sequences retain representations of choice options to support vicarious memory recall and deliberative decision-making, and function in a complementary manner with fast replay sequences involved in decision priming prior to onset of navigation, highlighting novel roles of these compressed sequences in guiding ongoing choice behavior. Notably, the mechanism of using internally generated sequences to simulate future scenarios has been used as a key feature that improves performance of model-based learning computations (*Daw and Dayan, 2014*; *Mattar and Daw, 2018*; *Pezzulo et al., 2019*; *Pezzulo et al., 2014*).

At the level of single theta cycles, we found that prior to the decision-making point in the navigation task, CA1 theta sequences report both actual and alternative choices, and are unable to predict chosen destination till after the decision has been made. On the other hand, we found robust PFC theta sequences that maintain prediction of upcoming choice prior to the decision point. Thus, theta sequences underlie a cognitive timescale mechanism that also maintains choice information, with the key distinction that this fast timescale mechanism can support vicarious memory recall of different choices, which was not seen at the behavioral timescale. These findings build on previous results that demonstrated theta-timescale activity patterns in the hippocampus representing future locations by recruiting cells encoding positions after the choice point (i.e. non-splitters; *Johnson and Redish, 2007*; *Kay et al., 2020*; *Papale et al., 2016*), and show that trajectory-specific coding of splitter cells at the behavioral timescale is preserved in fast theta sequences for representing future choices prior to the choice point as well. This process may be important in the event that animals have to change decisions or adapt to change in contingencies (*Buzsáki et al., 2014*; *Rich and Wallis, 2016*; *Yu and Frank, 2015*). In agreement with this idea, a recent study has shown that theta timescale mechanisms in CA1 can not only represent possible future paths, but also possible directions of motion on a moment-to-moment basis (*Kay et al., 2020*). Representation of past locations within theta cycles has also been recently reported (*Wang et al., 2020*). Complementing these results, previous studies have shown that theta oscillation cycles comprise cognitive computation units, corresponding to segregation of cell assemblies that represent different spatial experiences (*Brandon et al., 2013*; *Geisler et al., 2007*; *Gupta et al., 2012*; *Jezek et al., 2011*) and alternatives (*Johnson and Redish, 2007*; *Kay et al., 2020*; *Papale et al., 2016*). The results shown here establish that the representation of alternatives in the hippocampus interacts with prefrontal theta sequences in a content-specific manner, which can be used to guide actual choices.

The exact mechanism underlying these interactions, however, remains unclear. It has been shown that cortical neurons are sensitive to temporally organized inputs from the hippocampus (*Branco et al., 2010*; *Siapas et al., 2005*; *Sigurdsson et al., 2010*), and pharmacological disruption of prefrontal activity results in impaired theta sequences in the hippocampus (*Schmidt et al., 2019*), as well as impaired performance in the W-track task (*Maharjan et al., 2018*). Behavioral-state-dependent prefrontal-hippocampal interactions during theta oscillations can potentially also be

mediated through connections with entorhinal cortex (*Fernández-Ruiz et al., 2017*), nucleus reuniens (*Ito et al., 2015*), or other regions (*Eichenbaum, 2017*) that contribute to theta generation and memory retrieval. The specific roles of these circuits will need to be further elucidated in future investigations.

Notably, theta-associated mechanisms at both behavioral and theta timescales during active navigation supported retention during working memory periods, but did not predict errors in decisions. Rather, we found that compressed SWR replay sequences during inter-trial periods prior to the onset of trajectories prime the decision without apparent external cue triggers, which is maintained from the onset of the trajectory via behavioral and theta sequences. Previous studies have shown that SWR-reactivation in the hippocampus is less coordinated prior to incorrect versus correct trials (*Shin et al., 2019*; *Singer et al., 2013*), and disrupting awake SWRs leads to increase in errors in the spatial working memory task (*Fernández-Ruiz et al., 2019*; *Jadhav et al., 2012*). In addition, we have previously shown a relationship between reverse and forward CA1-PFC replay with past and future trajectories, respectively (*Shin et al., 2019*). Consistent with all these prior results, our current findings provide definitive evidence that coherent CA1-PFC replay of future trajectories prior to navigation onset primes the chosen destination in this memory-guided decision-making task, and errors in CA1-PFC replay predict incorrect decisions.

It is important to note that the expression of behavioral- and compressed-timescale sequences representing different trajectories are inextricably linked. Choice-specific representations of both theta and replay sequences depend on choice encoding through trajectory preferred firing of behavioral sequences. Furthermore, trajectory-selective neurons showed an extended tail of their spatial fields, which is inherently a behavioral-timescale characteristic, and likely contributes to trajectory-modulated look-ahead of theta sequences. Finally, there is evidence that degradation of theta sequences results in impaired sequential activation during sleep replay in the hippocampus (*Drieu et al., 2018*). Thus, it is the interactions among these multi-timescale activity patterns that potentially govern decision-making. The network mechanisms that enable expression of sequences at distinct timescales in multiple circuits remain a key question for future investigation.

Overall, our results provide a critical extension to classic models, which emphasize behavioral-timescale activity patterns typically spanning entire retention intervals, by establishing a role of discrete, fast timescale ensemble activity patterns in decision-making processes. Such a mechanism is broadly supported by recent findings of rapid shifts in activity patterns during decision-making (*Bernacchia et al., 2011*; *Durstewitz et al., 2010*; *Karlsson et al., 2012*; *Latimer et al., 2015*; *Rich and Wallis, 2016*; *Sadacca et al., 2016*), including discrete gamma oscillatory bursts in PFC underlying working memory (*Lundqvist et al., 2016*; *Miller et al., 2018*), which occur at a similar timescale to compressed theta and replay sequences. Together, these results suggest the possibility of transient LFP oscillations as informative signatures of fast evolving cell assemblies that bear on decision-making processes, and the cooperative behavioral- and cognitive-timescale mechanisms described here may reflect a general organizing principle of neural dynamics underlying decision-making.

# Materials and methods

**Key resources table**

| Reagent type (species) or resource | Designation | Source or reference | Identifiers | Additional information |
|---|---|---|---|---|
| Strain, strain background (*Long Evans rats; male*) | Long Evans | Charles River | Cat#: Crl:LE 006 RRID: RGD_2308852 | |
| Chemical compound, drug | Cresyl Violet | Acros Organics | Cat#: AC229630050 | |
| Chemical compound, drug | Formaldehyde | Fisher | Cat#: 50-00-0,67561, 7732-18-5 | |
| Chemical compound, drug | Isoflurane | Patterson Veterinary | Cat#: 07-806-3204 | |

*Continued on next page*

*Continued*

| Reagent type (species) or resource | Designation | Source or reference | Identifiers | Additional information |
|---|---|---|---|---|
| Chemical compound, drug | Ketamine | Patterson Veterinary | Cat#: 07-803-6637 | |
| Chemical compound, drug | Xylazine | Patterson Veterinary | Cat#: 07-808-1947 | |
| Chemical compound, drug | Atropine | Patterson Veterinary | Cat#: 07-869-6061 | |
| Chemical compound, drug | Bupivacaine | Patterson Veterinary | Cat#: 07-890-4881 | |
| Chemical compound, drug | Beuthanasia-D | Patterson Veterinary | Cat#: 07-807-3963 | |
| Software, algorithm | MATLAB 2017a | Mathworks, MA | RRID: SCR_001622, V2017a | |
| Software, algorithm | Trodes | SpikeGadgets | https://spikegadgets.com/trodes/, V1.9 | |
| Software, algorithm | Matclust | Mattias P. Karlsson | https://www.mathworks.com/matlabcentral/fileexchange/39663-matclust, V1.7 | |
| Software, algorithm | Libsvm | *Chang and Lin, 2011* | RRID:SCR_010243 https://www.csie.ntu.edu.tw/~cjlin/libsvm/, V3.12 | |
| Software, algorithm | Chronux | Partha Mitra | RRID:SCR_00554 http://chronux.org/, V2.12 | |
| Software, algorithm | measure_phaseprec Toolbox | *Kempter et al., 2012*; *Sanders et al., 2019* | https://github.com/HoniSanders/measure_phaseprec | |
| Software, algorithm | Prism 8 | GraphPad Software | RRID: SCR_002798, V8.0 | |

## Subjects

Nine adult male Long-Evans rats (450–550 g, 4–6 months) were used in this study. All procedures were approved by the Institutional Animal Care and Use Committee at the Brandeis University and conformed to US National Institutes of Health guidelines. Data from six subjects have been reported in an earlier study (*Shin et al., 2019*).

## Animal pre-training

Animals were habituated to daily handling for several weeks before training. After habituation, animals were food deprived to 85–90% of their ad libitum weight, and pre-trained to run on a linear track (~1 m long) for rewards (sweetened evaporated milk), and habituated to an high-walled, opaque sleep box (~30 × 30 cm) as described previously (*Jadhav et al., 2012*; *Jadhav et al., 2016*; *Tang et al., 2017*). After the pre-training, animals were surgically implanted with a multi-tetrode drive.

## Surgical implantation

Surgical implantation procedures were as previously described (*Jadhav et al., 2012*; *Jadhav et al., 2016*; *Shin et al., 2019*; *Tang et al., 2017*). Eight animals were implanted with a multi-tetrode drive containing 32 independently moveable tetrodes targeting right dorsal hippocampal region CA1 (−3.6 mm AP and 2.2 mm ML) and right PFC (+3.0 mm AP and 0.7 mm ML) (16 tetrodes in CA1 and 16 in PFC for four animals; 13 in CA1 and 19 in PFC for three animals; 15 in CA1 and 17 in PFC for one animal). One animal was implanted with a multi-tetrode drive containing 64 independently moveable tetrodes targeting the bilateral CA1 of dorsal hippocampus (−3.6 mm AP and ±2.2 mm ML; *Figure 1—figure supplement 1A, left*) and PFC (+3.0 mm AP and ±0.7 mm ML; *Figure 1—figure supplement 1B, left*) (30 tetrodes in CA1 and 34 tetrodes in PFC). On the days following surgery, hippocampal tetrodes were gradually advanced to the desired depths with characteristic EEG patterns (sharp wave polarity, theta modulation) and neural firing patterns as previously described (*Jadhav et al., 2012*; *Jadhav et al., 2016*; *Shin et al., 2019*; *Tang et al., 2017*). One tetrode in corpus callosum served as hippocampal reference (CA1 REF), and another tetrode in overlying cortical

regions with no spiking signal served as prefrontal reference (PFC REF). The reference tetrodes reported voltage relative to a ground (GND) screw installed in skull overlying cerebellum. Electrodes were not moved at least 4 hr before and during the recording day.

## Behavioral task

Following recovery from surgical implantation (~7–8 days), animals were food-deprived and again pre-trained on a linear track for at least 2 days before the W-track sessions started. During the recording day, animals were introduced to the novel W-track (*Figure 1A*;~80 × 80 cm with ~7 cm wide tracks) for the first time, and learned the task rules over eight behavioral sessions (or epochs, denoted as E1-E8; *Figure 1B*). Each behavioral session lasted 15–20 min and was interleaved with 20–30 min rest sessions in the sleep box (total recording duration $\cong$ 6 hr within a single day) (*Shin et al., 2019*). On the W-maze, animals were rewarded for performing a hippocampus- (*Fernández-Ruiz et al., 2019*; *Jadhav et al., 2012*; *Kim and Frank, 2009*) and prefrontal-dependent (*Maharjan et al., 2018*) continuous alternation task: returning to the center well after visits to either side well (left or right well; inbound trajectories), and choosing the opposite side well from the previously visited side well when starting from the center well (outbound trajectories). Rewards were automatically delivered in the reward wells (left well: L; right well: R; center well: C) triggered by crossing of an infrared beam by the animal's nose. Therefore, animals performed four types of trajectories during correct behavioral sequences in this task: center-to-left (C-to-L), left-to-center (L-to-C), center-to-right (C-to-R), and right-to-center (R-to-C). Among these trajectory types, C-to-L and C-to-R are outbound trajectories, while L-to-C and R-to-C are inbound trajectories. When animals were on the center stem, the two inbound trajectories thus represented possible past paths (one actual, and one alternative; *Figure 1A*, *left*), and the two outbound trajectories represented possible future paths (*Figure 1A*, *right*). The learning curves were estimated using a state-space model (*Figure 1B*; *Jadhav et al., 2012*; *Shin et al., 2019*; *Smith et al., 2004*).

## Behavioral analysis

Locomotor periods, or theta states, were defined as periods with running speed >5 cm/s, whereas immobility was defined as periods with speed ≤4 cm/s. The animal's arrival and departure at a reward well was detected by an infrared beam triggered at the well. The well entry was further refined as the first time point when the speed fell below 4 cm/s before the arrival trigger, whereas the well exit was defined as the first time point when the speed rose above 4 cm/s after the departure trigger (*Shin et al., 2019*). The time spent at a reward well (i.e. immobility period at well) was defined as the period between well entry and exit. Only SWRs occurring during immobility periods at reward wells were analyzed in this study (see also *SWR detection*). The center stem of the W-maze was defined as the set of linear positions (see *Spatial firing rate maps and linearization*) between the center well and the center junction (i.e., choice point, CP). For a given behavioral trajectory, the before-CP period was defined as the time spent at the center stem, and the after-CP period was defined as the time spent at locations between 10 cm away from the center stem and the side well (*Figure 6* and *Figure 6—figure supplements 1–2*). Therefore, for outbound trajectories, before-CP periods began when animals exited the center well and ended when animals reached the choice point (*Figure 6*), and for inbound trajectories, before-CP periods began when animals entered the choice point from the side arm and ended when animals entered the center well (*Figure 6—figure supplements 1–2*).

## Neural recordings

Data were collected using a SpikeGadgets data acquisition system (SpikeGadgets LLC) (*Shin et al., 2019*; *Tang et al., 2017*). Spike data were sampled at 30 kHz and bandpass filtered between 600 Hz and 6 kHz. LFPs were sampled at 1.5 kHz and bandpass filtered between 0.5 Hz and 400 Hz. The animal's position and running speed were recorded with an overhead color CCD camera (30 fps) and tracked by color LEDs affixed to the headstage. Single units were identified by manual clustering based on peak and trough amplitude, principal components, and spike width using custom software (MatClust, M. P. Karlsson) as previously described (*Jadhav et al., 2016*; *Shin et al., 2019*; *Tang et al., 2017*). Only well isolated neurons with stable spiking waveforms were included (*Shin et al., 2019*).

## Histology

Following the conclusion of the experiments, micro-lesions were made through each electrode tip to mark recording locations (*Shin et al., 2019*). After 12–24 hr, animals were euthanized (Beuthanasia) and intracardially perfused with 4% formaldehyde using approved procedures. Brains were fixed for 24 hr, cryoprotected (30% sucrose in 4% formaldehyde), and stored at 4°C. The recording sites were determined from *post hoc* Nissl-stained coronal brain sections based on *The Rat Brain in Stereotaxic Coordinates* (*Paxinos and Watson, 2004*; *Figure 1—figure supplement 1A–B*).

## Cell inclusion

Units included in analyses fired at least 100 spikes in a given session. Putative interneurons were identified and excluded based on spike width and firing rate criterion as previously described (*Jadhav et al., 2012*; *Jadhav et al., 2016*). Peak rate for each unit was defined as the maximum rate across all spatial bins in the linearized spatial map (see *Spatial firing rate maps and linearization*). A peak rate $\geq$3 Hz was required for a cell to be considered as a place cell.

## Spatial firing rate maps and linearization

Spatial firing rate maps (or rate maps) were calculated only during locomotor periods (> 5 cm/s; all SWR times excluded) at positions with sufficient occupancy (> 20 ms). To construct the 1D linearized spatial firing rate maps on different trajectory types, animal's linear positions were first estimated by projecting its actual 2D positions onto pre-defined idealized paths along the track, and further classified as belonging to one of the four trajectory types (*Shin et al., 2019*; *Tang et al., 2017*). The linearized spatial firing rate maps were then calculated using spike counts and occupancies in 2-cm bins of the linearized positions and smoothened with a Gaussian curve (4-cm SD). We found all linearized positions along each trajectory type were sufficiently covered by the spatial firing rate maps of CA1, as well as PFC, populations (*Shin et al., 2019*).

## Trajectory selective index

To measure the trajectory selectivity of single cells, a trajectory selectivity index (*SI*) was calculated by comparing the mean firing rates on the Left- (or L-) vs. Right- (or R-) side trajectories for outbound (C-to-L vs. C-to-R) and inbound (L-to-C vs. R-to-C), respectively:

$$SI = \frac{FR_L - FR_R}{FR_L + FR_R},$$

where $FR_L$ is the mean firing rate on the L-side trajectory, and $FR_R$ is for the R-side trajectory. Only cells that had a peak firing rate $\geq$3 Hz detected on either the L- or R-side trajectory were considered, and the rate maps in different sessions were analyzed separately. A cell with $|SI| > 0.4$ in CA1, or $|SI| > 0.2$ in PFC was classified as trajectory-selective cells ($|SI| = 0.384 \pm 0.004$ and $0.154 \pm 0.003$ for all CA1 and PFC cells, respectively; mean $\pm$ SEM) (*Kay et al., 2020*). The trajectory type (L vs. R) with highest firing rate was designated as the cell's preferred (Pref) trajectory, and the other type designated as the non-preferred (Non-pref) trajectory.

## SWR detection

Sharp-wave ripples (SWRs) were detected during immobility periods ($\leq$ 4 cm/s) as described previously (*Jadhav et al., 2012*; *Jadhav et al., 2016*; *Shin et al., 2019*; *Tang et al., 2017*). In brief, LFPs from CA1 tetrodes relative to the CA1 reference tetrode were filtered into the ripple band (150-250 Hz), and the envelope of the ripple-filtered LFPs was determined using a Hilbert transform. SWRs were initially detected as contiguous periods when the envelope stayed above 3 SD of the mean on at least one tetrode, and further refined as times around the initially detected events during which the envelope exceeded the mean. For replay and reactivation analysis, only SWRs with a duration $\geq$ 50 ms were included as in previous studies (*Pfeiffer and Foster, 2013*; *Shin et al., 2019*).

## Theta phases and theta cycles

Peaks and troughs of theta oscillations, as well as theta phases, were identified on the band-passed (6-12 Hz) LFPs from the CA1 reference tetrode (CA1 REF) (*Jadhav et al., 2016*; *Lubenov and Siapas, 2009*). To precisely define a theta cycle for theta sequence detection, theta phase locking of

each cell in CA1 was calculated across locomotor periods (> 5 cm/s) in each session using the methods developed in previous reports (*Jadhav et al., 2016*; *Siapas et al., 2005*). A phase histogram was then calculated by averaging across all phase-locked CA1 cells (Rayleigh tests at p < 0.05) in each session, and the phase with minimum cell firing was used to separate theta cycles in the given session (approximately valley-to-valley of hippocampal REF theta, or peak-to-peak of hippocampal fissure theta) (*Gupta et al., 2012*).

## Theta phase precession

Theta phase precession was examined in linearized spatial firing rate maps with a peak rate $\geq$ 3 Hz, and multiple fields of a single cell were analyzed separately (e.g. *Figure 4E*, *top right*). For each firing peak of linearized spatial firing rate maps detected (using MATLAB *findpeaks* function with a 20-cm minimal peak distance), a spatial field was defined as contiguous positions with rate > 10% of the peak rate, and at least 8 cm large (*Figure 4E and F*; *Schmidt et al., 2009*). For spikes within each spatial field, phase precession was computed using a circular-linear fit as previously described (*cl_corr* function in the *measure_phaseprec* toolbox; https://github.com/HoniSanders/measure_phaseprec) (*Kempter et al., 2012*; *Sanders et al., 2019*). The slope, correlation coefficient (*r*), and its *p*-value from the circular-linear regression were reported (*Figure 4E-H*).

## Theta power and coherence

Power spectra and coherograms were computed from the LFPs referenced to GND using multitaper estimation methods from the Chronux toolbox (http://chronux.org; version 2.12) (*Shin et al., 2019*). We obtained the SD and mean for each frequency across a given session, and normalized the power of that frequency as a z-score (*Figure 1—figure supplement 1B*). Coherence between a pair of CA1 and PFC tetrodes was calculated during locomotor periods (> 5 cm/s; locations within 15 cm of the reward well were excluded to prevent contamination from SWR activity). Coherograms averaged over all available CA1-PFC tetrode pairs with simultaneously recorded LFPs were shown in *Figure 1—figure supplement 1C-D*. Theta power and coherence were measured as the mean power and coherence between 6-12 Hz, respectively.

## Spatial field asymmetry

For cells showing significant phase precession (circular-linear regression at p < 0.05), we further analyzed their spatial field asymmetry (*Figure 4A-D*). Only fields with the highest peak rate of a single cell for each trajectory type, and at least 20 cm large, were used. The spatial fields were then binned into 10% field length relative to field center (*Souza and Tort, 2017*), and the asymmetry index (*AI*) was calculated as:

$$AI = \frac{A_R - A_L}{A_R + A_L},$$

where $A_R$ denoted the area under the firing rate profile to the right of the field center (i.e. x > 0 in *Figure 4A and B*), while $A_L$ represented the same to the left of the field center (i.e. x < 0 in *Figure 4A and B*). Therefore, a negative *AI* corresponds to a spatial field with an extended initial tail.

## Theta cycle skipping

To quantify theta cycle skipping in single cells (*Figure 5*), we measured a cycle skipping index (CSI) on their auto-correlograms (ACGs). Data on different trajectory types were analyzed separately, and thus a single cell could contribute to more than one ACG (*Figure 5B–D*). For each ACG, data was restricted to locomotor periods (>5 cm/s) that lasted at least 1.5 s, and with at least total 100 spikes (*Kay et al., 2020*). Each ACG was first estimated as a histogram of nonzero lags across the interval ±400 ms (bin = 10 ms; denoted as *ACG_raw*) (*Brandon et al., 2013*; *Kay et al., 2020*), and was further corrected for the triangular shape caused by finite duration data (*Kay et al., 2020*; *Mizuseki et al., 2009*):

$$ACG(t) = \frac{ACG\_raw(t)}{1 - \frac{|t|}{T}},$$

where $t$ is the time lag (-$T < t < T$), and $T$ is the total duration of the spike train used to compute the ACG. The corrected ACG was then smoothed (Gaussian kernel, SD = 20 ms) and peak-normalized. To detect the theta-modulated peaks of ACGs, power spectra for ACGs were generated using FFT, and the relative theta power of an ACG was calculated by dividing its power in the theta band (6-10 Hz) by its total power in the 1-50 Hz range (*Deshmukh et al., 2010*). An ACG with relative theta power > 0.15 was considered theta-modulated. For all theta-modulated ACGs, the ACGs were band-pass filtered between 1 and 10 Hz (*Deshmukh et al., 2010*), and the amplitudes of the first and second theta peaks on the filtered ACG were then determined by finding a first peak ($p_1$) near $t$ = 0 in the 90-200 ms window, and the second peak ($p_2$) near $t$ = 0 in the 200-400 ms window, as described previously (*Kay et al., 2020*). The CSI was then determined as:

$$CSI = \frac{p_2 - p_1}{max(p_1, p_2)},$$

The CSI ranges between −1 and 1, and higher values indicate more theta cycle skipping.

## Sequence analysis

Sequence analysis here focused on three different ensemble sequences: behavioral sequences, theta sequences, and replay sequences (*Figure 1C*). To evaluate neural representations at the ensemble level, Bayesian decoding was implemented as previously described (*Davidson et al., 2009*; *Shin et al., 2019*; *Tang et al., 2017*; *Zhang et al., 1998*): a memoryless Bayesian decoder was built for different trajectory types (for outbound, C-to-L vs. C-to-R; for inbound, L-to-C vs. R-to-C) to estimate the probability of animals' position given the observed spikes (Bayesian reconstruction; or posterior probability matrix):

$$P(X, Tr|\mathbf{spikes}) = \frac{P(\mathbf{spikes}|X, Tr)P(X, Tr)}{P(\mathbf{spikes})},$$

where $X$ is the set of all linear positions on the track for different trajectory types (i.e., $Tr$; $Tr \in \{L, R\}$, where $L$ represents the L-side trajectory, $R$ represents the R-side trajectory), and we assumed a uniform prior probability over $X$ and $Tr$. Assuming that all $N$ cells active in a sequence fired independently and followed a Poisson process:

$$P(\mathbf{spikes}|X, Tr) = \prod_{i=1}^{N} P(spikes_i|X, Tr) = \prod_{i=1}^{N} \frac{(\tau f_i(X, Tr))^{spikes_i} e^{-\tau f_i(X, Tr)}}{spikes_i!},$$

where $\tau$ is the duration of the time window (see below), $f_i(X, Tr)$ is the expected firing rate of the $i$-th cell as a function of sampled location $X$ and trajectory type $Tr$, and $spikes_i$ is the number of spikes of the $i$-th cell in a given time window. Therefore, the posterior probability matrix can be derived as follows:

$$P(X, Tr|\mathbf{spikes}) = C\left(\prod_{i=1}^{N} f_i(X, Tr)^{spikes_i}\right) e^{-\tau \sum_{i=1}^{N} f_i(X, Tr)},$$

where $C$ is a normalization constant such that $\sum_{k=1}^{2} \sum_{j=1}^{D} P(x_j, tr_k|\mathbf{spikes}) = 1$ ($x_j$ is the $j$-th position bin, $D$ is the total length of the track, and $tr_k$ is the $k$-th trajectory type; $k$ = 1 or 2, representing L- or R-side trajectory, respectively).

Specifically, for behavioral sequences, the Bayesian decoder was used to decode animal's current location ($x$) and choice ($tr$) (*Figure 1F–I* and *Figure 1—figure supplement 1C–H*) as in previous studies (*Shin et al., 2019*). Data was restricted to locomotor periods (>5 cm/s; locations within 15 cm of the reward well were excluded for decoding to prevent contamination from SWR activity), and binned into 120 ms bins (i.e. $\tau$ = 120 ms; moving window with 60 ms overlap). For each time bin, the location and choice (i.e. trajectory type) with maximum decoded probability was compared to the actual position and choice of the animal in that bin (*Figure 1F–I* and *Figure 1—figure supplement 1C–H*). Decoding error of positions in this bin was determined as the linear distance between estimated position and actual position (*Figure 1—figure supplement 1C–H*), and the accuracy of animal's choices decoded was reported (*Figure 1H and I*).

For theta sequences, we first defined candidate events as theta cycles with at least five cells active in a given brain region (CA1 or PFC). Only theta cycles with running speed >10 cm/s, and a duration ranging from 100 to 200 ms were used (*Feng et al., 2015*). A time window of 20 ms (i.e. $\tau$ = 20 ms; moving window with 10 ms overlap) was used to examine theta sequence structure at a fast, compressed timescale.

To calculate the distance index of each candidate event (*Figure 3C and D*), decoded probabilities ± 60 cm around the animal's current location, and ±1/2 cycle around the 0 phase of the theta cycle (i.e. -π to 0 as 1st half of theta phases, and 0 to π as 2nd half of theta phases), were divided equally into four quadrants (*Feng et al., 2015*). The distance index of the 1st half of theta phases is thus measured by comparing the probabilities in the quadrants ahead (quadrant III, future) and behind (quadrant II, past) as (III – II)/ (III + II). Similarly, the distance index of the 2nd half of theta phases is thus measured as (IV – I)/ (IV + I).

To identify sequential structure within a theta cycle, two measures were adapted from previous theta-sequence studies (*Drieu et al., 2018*; *Farooq and Dragoi, 2019*; *Feng et al., 2015*; *Zheng et al., 2016*). In the first method, a weighted correlation (*Farooq and Dragoi, 2019*; *Feng et al., 2015*), $r(x,t|\mathbf{Pmat})$, was calculated for the posterior probability matrix of each trajectory type (**Pmat**, $D \times T$, $D$ is the total number of spatial bins, and $T$ is the total number of temporal bins). The weighted means were computed across locations ($x$) and time ($t$) as:

$$E_X(x|\mathbf{Pmat}) = \frac{\sum_{i=1}^{T}\sum_{j=1}^{D} Pmat_{ij}x_j}{\sum_{i=1}^{T}\sum_{j=1}^{D} Pmat_{ij}},$$

$$E_T(t|\mathbf{Pmat}) = \frac{\sum_{j=1}^{D}\sum_{i=1}^{T} Pmat_{ij}t_i}{\sum_{j=1}^{D}\sum_{i=1}^{T} Pmat_{ij}},$$

and the weighted covariance, $covar(x,t|\mathbf{Pmat})$, was computed as:

$$covar(x,t|\mathbf{Pmat}) = \frac{\sum_{i=1}^{T}\sum_{j=1}^{D} Pmat_{ij}\left(x_j - E_X(x|\mathbf{Pmat})\right)\left(t_i - E_T(t|\mathbf{Pmat})\right)}{\sum_{i=1}^{T}\sum_{j=1}^{D} Pmat_{ij}},$$

where $t_i$ is the $i$-th temporal bin, and $x_j$ is the $j$-th spatial bin of the posterior probability matrix (**Pmat**). The weighted correlation was then calculated as:

$$r(x,t|\mathbf{Pmat}) = \frac{covar(x,t|\mathbf{Pmat})}{\sqrt{covar(t,t|\mathbf{Pmat})covar(x,x|\mathbf{Pmat})}},$$

and the weighted correlation was reported as the sequence score ($r$).

In the second method, we measured whether the decoded positions in successive temporal bins of the posterior probability matrix were tightly arranged along an oblique line as previously reported (*Davidson et al., 2009*; *Drieu et al., 2018*; *Feng et al., 2015*). Briefly, the best-fit line of a theta sequence (e.g. yellow lines of the decoding plots in *Figure 2A–F*) was determined by a fitted line that yielded maximum posterior probability in an 8 cm vicinity ($d$). For a given candidate line with a slope $v$ and an intercept $\rho$, the average likelihood $R$ that the decoded position is located within a distance $d$ of that line is:

$$R(v,\rho) = \frac{1}{n}\sum_{k=0}^{n-1} P(|pos - vk \cdot \Delta t| \le d),$$

where $k$ is the temporal bin of the posterior probability matrix, and $\Delta t$ is the moving step of the decoding window (i.e. 10 ms). To determine the best-fit line for each theta sequence, we densely sampled the parameter space of $v$ and $\rho$ ($v > 1$ m/s to exclude stationary events) to find the value that maximized $R$ ($R_{max}$, i.e. goodness-of-fit).

In order to assess the significance of theta sequences, we circularly shifted the space-bins of the posterior probability matrix ($n$ = 1000 times) as described previously (*Drieu et al., 2018*; *Farooq and Dragoi, 2019*; *Zheng et al., 2016*), and calculated the weighted correlation and the goodness-of-fit from the shuffled data. A sequence was considered significant if it met two criteria:

first, its sequence score (i.e. weighted correlation) exceeded the 97.5th percentile or was below the 2.5th percentile (for reverse sequences) of the shuffled distributions; and its goodness-of-fit ($R_{max}$) was higher than the 95th percentile of their shuffles. We considered the significant trajectory type as the decoded trajectory, and if more than one trajectory type were significant, the trajectory with the highest sequence score was considered as the decoded trajectory. For plotting purposes only, a moving window (30 ms, advanced in steps of 5 ms) was used for displaying theta sequences (*Figures 2* and *6* and *Figure 2—figure supplement 1* and *Figure 6—figure supplement 1*).

For synchronous CA1 and PFC theta sequences, we examined if their representations were coherent or independent (*Figure 6J-L*). If the representations of CA1 and PFC theta sequences (denoted as $Seq_{CA1}$ and $Seq_{PFC}$, respectively) are stochastically independent, then $p(Seq_{PFC} = $ actual$| Seq_{CA1} = $ actual$) = p(Seq_{PFC} = $ actual$| Seq_{CA1} = $ alternative$) = p(Seq_{PFC} = $ actual$) = 1 - p(Seq_{PFC} = $ alternative$)$. Given that $p(Seq_{PFC} = $ actual$)$ and $p(Seq_{PFC} = $ alternative$)$ are significantly different from the chance level (50%; p's < 1e-4, signed-rank tests compared to 50%), stochastic independence would predict that the distributions of sequences coherently representing actual and alternative (i.e., $p(Seq_{PFC} = $ actual$| Seq_{CA1} = $ actual$)$ and $p(Seq_{PFC} = $ alternative $| Seq_{CA1} = $ alternative$)$) both significantly differ from the chance level. However, if they are dependent, different distributions for actual and alternative should be observed (*Figure 6L*).

The detection of replay sequences has been described previously (*Shin et al., 2019*). Briefly, candidate replay events were defined as the SWR events during which $\geq$ 5 place cells fired. Each candidate event was then divided into 10 ms non-overlapping bins (i.e. $\tau$ = 10 ms), and decoded based on the Bayes' rule described above. The assessment of significance for replay events was implemented by a Monte Carlo shuffle, in which the *R*-squared from linear regression on the temporal bins versus the locations of the posterior probability matrix was compared to the *R*-squared derived from the shuffled data (i.e. time shuffle, circularly shuffling temporal bins of the posterior probability matrix). A candidate event with an *R*-squared that exceeded the 95th percentile of their shuffles (i.e. p<0.05) was considered as a replay event.

For SWR events with significant CA1 replay sequences detected, we further measured CA1-PFC reactivation during these events (*Figure 7—figure supplement 2*) as described previously (*Shin et al., 2019*). We only analyzed the events where $\geq$ 5 place cells and $\geq$5 PFC cells fired. Briefly, for an event with *N* CA1 and *M* PFC cells active, a (*N* × *M*) synchronization matrix during RUN ($\mathbf{C_{RUN}}$) was calculated with each element ($C_{i,j}$) representing the Pearson correlation coefficient of the linearized spatial firing rate maps on a certain trajectory type (see *Spatial firing rate maps and linearization*) of the *i*-th CA1 cell and the *j*-th PFC cell. To measure the population synchronization pattern during the SWR, the spike trains during the SWR were divided into 10 ms bins and z transformed (*Peyrache et al., 2009*; *Shin et al., 2019*). The (*N* × *M*) synchronization matrix during the SWR ($\mathbf{C_{SWR}}$) was then calculated with each element ($C_{i,j}$) representing the correlation of the spike trains of a CA1-PFC cell pair. Finally, the reactivation strength of this event for each trajectory type was measured as the correlation coefficient (*R*) between the population matrices, $\mathbf{C_{RUN}}$ and $\mathbf{C_{SWR}}$.

## Theta-sequence prediction of behavioral choices

For theta-sequence prediction of behavioral choices (*Figure 6H and I* and *Figure 6—figure supplement 2*), trial-by-trial classification analysis was performed using support vector machines (SVMs) through the *libsvm* library (version 3.12) (*Chang and Lin, 2011*). For each region (CA1 or PFC), two independent SVMs were trained for before-CP and after-CP periods. For each trial, the numbers of theta sequences representing the actual versus the alternative choice during a given period (before CP, or after CP) were used as a feature (*n* = 2) to predict the current choice (*k* = 2, L or R). All classifiers were *C*-SVMs with a radial basis function (Gaussian) kernel and trained on correct trials. Hyperparameter (*C* and $\gamma$; regularization weight and radial basis function width, respectively) selection was performed using a random search method with leave-one-out cross-validation to prevent overfitting. The selected hyperparameters were then used to report the leave-one-out cross-validation accuracy. The percentage of correctly inferred trials was computed across all training/test trial combinations to give prediction accuracy. The significance of this prediction was determined by comparing to the distribution of shuffled data by randomly shuffling the trial labels (L or R), and this shuffled dataset was used to train a classifier in the same way as the actual dataset. A prediction accuracy based on the actual dataset that was higher than the 95th percentile of its shuffles (p < 0.05) was considered

as significant. Only trials with at least one theta sequence for actual and alternative choices were used for prediction.

## CA1-PFC reactivation prediction of correct and incorrect responses

For prediction of correct and incorrect responses with CA1-PFC reactivation (*Figure 7F*), SVMs were used similar to the theta-sequence prediction analysis (see above). Two independent SVMs were trained on forward and reverse replay events. For each trial, the averaged CA1-PFC reactivation strength for the actual versus the alternative trajectory across all reactivation events during immobility at the reward well *prior* to the trial was used as a feature (n = 2; *Figure 7—figure supplement 2A-B*) to predict correct versus incorrect responses (k = 2, correct or incorrect, regardless of which side arm that the animals choose *Singer et al., 2013*). The significance of this prediction was determined by comparing it to the distribution of shuffled data by randomly shuffling the trial labels (correct or incorrect), and this shuffled dataset was used to train a classifier in the same way as the actual dataset. Given the unbalanced nature of the dataset (a lot more correct trials than incorrect trials), we resampled the incorrect trials (with replacement) to match the correct trials and used ROC analysis to measure the predictive power of the classifiers. The area under each ROC curve (AUC) was computed, and an AUC based on the actual dataset that was higher than the 95th percentile of its shuffles (p < 0.05) was considered as significant. Only trials with at least one reactivation event of the given type (forward or reverse) were used for prediction.

## Statistical analysis

Data analysis was performed using custom routines in MATLAB (MathWorks) and GraphPad Prism 8 (GraphPad Software). We used nonparametric and two-tailed tests for statistical comparisons throughout the paper, unless otherwise noted. We used Kruskal-Wallis or Friedman test for multiple comparisons, with post hoc analysis performed using a Dunn's test. p<0.05 was considered the cut-off for statistical significance. Unless otherwise noted, values and error bars in the text denote means ± SEMs. No statistical methods were used to pre-determine sample sizes, but our sample sizes are similar to those generally employed in the field.

## Acknowledgements

This work was supported by NIH Grant R01 MH112661 and the Smith Foundation Odyssey award to SPJ.

## Additional information

### Funding

| Funder | Grant reference number | Author |
| --- | --- | --- |
| National Institutes of Health | R01 MH112661 | Shantanu P Jadhav |
| Richard and Susan Smith Family Foundation | Odyssey award | Shantanu P Jadhav |

The funders had no role in study design, data collection and interpretation, or the decision to submit the work for publication.

### Author contributions

Wenbo Tang, Conceptualization, Data Curation, Software, Formal analysis, Validation, Visualization, Methodology, Writing - original draft, Writing - review and editing; Justin D Shin, Data curation, Validation, Investigation, Methodology, Writing - original draft, Writing - review and editing; Shantanu P Jadhav, Conceptualization, Supervision, Project administration, Funding acquisition, Methodology, Writing - original draft, Writing - review and editing

## Author ORCIDs

Wenbo Tang (iD) https://orcid.org/0000-0003-4361-6705
Justin D Shin (iD) https://orcid.org/0000-0002-7959-7772
Shantanu P Jadhav (iD) https://orcid.org/0000-0001-5821-0551

## Ethics

Animal experimentation: All procedures were approved by the Institutional Animal Care and Use Committee at Brandeis University (protocol #21001) and conformed to US National Institutes of Health guidelines.

## Decision letter and Author response

Decision letter https://doi.org/10.7554/eLife.66227.sa1
Author response https://doi.org/10.7554/eLife.66227.sa2

## Additional files

### Supplementary files

• Transparent reporting form

### Data availability

All data generated or analysed during this study are included in the manuscript and source data files.

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
