## [Decision Letter]

**Acceptance summary:**

This is an exciting and important manuscript that will be of broad interest to readers in the field of learning and memory, place cells, sharp wave-ripples, and theta rhythms, as well as those interested in the functional significance of hippocampal-medial prefrontal cortex networks. This paper involves recordings of large neuronal ensembles in the hippocampus and medial prefrontal cortex in animals performing a hippocampal-dependent spatial working memory task. The paper provides a number of descriptions, at different timescales and with different analyses, of how neural activity in the rat hippocampus and prefrontal cortex relates to behavioral performance of a memory task.

**Decision letter after peer review:**

Thank you for submitting your article "Multiple time-scales of decision making in the hippocampus and prefrontal cortex" for consideration by *eLife*. Your article has been reviewed by three peer reviewers, one of whom is a member of our Board of Reviewing Editors, and the evaluation has been overseen by Laura Colgin as the Senior and Reviewing Editor. The reviewers have opted to remain anonymous.

Essential Revisions:

Reviewers agreed that this is a strong submission with beautiful data and figures, as well as analyses that are appropriate and solid. However, they felt that there was a lack of clarity regarding the paper's major conclusions and how some of the analyses led to those conclusions. Reviewers are confident that the authors will be able to address these concerns. No new data are required to support the conclusions.

Reviewer #1 (Recommendations for the authors (required)):

1) To avoid confusion, the authors should perhaps reconsider using the phrase "population spikes" here. This phrase often has a different meaning in neurophysiology from what is meant here.

2) To improve readability, the authors may want to consider not using the needless, non-standard abbreviation CP for choice point.

3) In the figures, perhaps a different color than blue could be used for the left arm to improve clarity and readability. The color blue indicates outbound in some figures/panels and left arm trajectory in other figures/panels.

4) I am confused by the broken y-axis in Figure 1B. Why is the axis disrupted with 0.4 written twice?

5) The distance index in Figure 3C-D is not described in the Materials and methods section. The description in the figure legend is confusing. The description states that it compares posterior probabilities for future versus past locations, making it sound as though this means future and past relative to the present location of the animal. However, the figure legend also refers to theta phases, making it sound as though this measure was calculated based on theta phases of spikes. How exactly was this measure computed?

6) The claim "Note the robust difference of CA1-PFC reactivation of taken versus not-taken path during correct trials compared to that during incorrect trials" is not well supported because no interaction effect is reported (i.e., interaction showing that CA1-PFC reactivation strength for same vs. alternative paths was significantly different for correct vs. incorrect trials). It seems as if only multiple paired t-tests were used here. Also, the data for correct and incorrect trials are plotted on different y-axis scales for panels A and B in Figure 7—figure supplement 2, making comparisons of effects difficult.

Reviewer #2 (Recommendations for the authors (required)):

In this manuscript, Tang and colleagues extend previous work on medial prefrontal-hippocampal interactions in several important ways. In rats performing a well-validated spatial working memory task, they demonstrate the existence of theta sequences in PFC ensembles that mirror the organizational properties of theta sequences in hippocampus (albeit with less reliability than the canonical spatial representations in hippocampus). They show further that PFC theta sequences are coordinated with theta sequence representations in hippocampus; specifically, when hippocampus represents the animal's impending choice destination in a theta cycle, there is an increased likelihood that the simultaneous PFC representation shows the same representation. Finally, the authors show that the content of these fast timescale "cognitive" representations in theta cycles is dissociable from slower "behavioral timescale" representations. Specifically, at the behavioral time scale so called "splitter cells" provide sufficient information to decode impending choice from either hippocampus or PFC. In contrast, fast time scale representations diverged between the two structures before the animal committed to one choice option, with PFC representing the animals upcoming choice and hippocampus representing both possible choice options. This dissociation, however, was dynamically modulated, because after animals committed to a decision, both structure represented the chosen outcome.

I think this manuscript is a potentially important contribution to our understanding of PFC-hippocampal dynamics. First, the identification of clear theta sequences in PFC is, on its own, an important result that is complementary to a recent report showing reactivation of spatially-tuned PFC cells that is highly reminiscent of reactivation of place cells in hippocampus (Kaefer...Csicsvari, 2020). In addition, however, the authors probe the coordination of prefrontal and hippocampal theta sequences, and find that there is an apparent relationship that is dynamically modulated by the behavioral demands of the decision making task. Together, I think the body of work reported here is an important contribution to our understanding of how these structures work together.

The analysis is described clearly and with excellent detail, and appears rigorous and appropriate to the authors' questions. This is an incredibly information-rich manuscript, but it is well organized and well explained throughout. Overall, I think this is a strong contribution, and I have only very minor suggestions to improve it.

At several points the authors mention that results in the present manuscript replicated their previous report (Shin et al 2019). This is important, but is somewhat tempered by the fact that 6/9 rats were reanalyzed from that 2019 data set.

Reviewer #3 (Recommendations for the authors (required)):

This is an unusual paper in that it covers a large amount of ground, with 4-5 main claims that each individually have been (or could be) the subject of entire papers. In its attempt to integrate each of these findings into an overall theory, the paper reads a bit like a hybrid between a review and an empirical paper. One the one hand, this is a strength because it aspires to much-needed synthesis; on the other hand, I found it difficult to identify which statements are the primary empirical claims advanced in the paper, and which statements are interpretations. Moreover, for those statements that looked like main claims, I am not convinced that as presented, there is sufficient support. Thus, overall, I think the paper requires (a) more explicit statements about what the key primary claims are, and (b) additional analyses to better support those claims and/or better separation of result and interpretation.

To expand on the above: as far as I can tell, the main novel claims seem to be:

1) the prefrontal cortex (PFC) expresses theta sequences

2) theta sequences in both the hippocampus (HC) and PFC are trajectory-specific ("splitter sequences")

3) PFC, but not HC, theta sequence content predicts future choice

4) joint HC-PFC activity during replay predicts error vs. correct trials

Based on these findings, the authors conclude that interaction at multiple timescales between HC and PFC supports memory-guided decision-making.

If adequately supported, each of these claims would be a useful addition to the field, with the dissociation between HC and PFC theta sequences of particular interest. However, as currently presented, the claims seem out of step with the underlying evidence; in each case, this overall issue could be addressed by bolstering the analyses and/or by making the claims more precise or toned down.

Specifically:

Claim 1: What counts as a "theta sequence" requires clarification. Do the authors consider the ensemble pattern generated from independently phase precessing cells a theta sequence? If yes, then the claim that mPFC has theta sequences follows directly from previous single cell results (Jones and Wilson, 2005 PLoS Biology; Zielinski et al., 2019 J Neurosci) and they should acknowledge this. If no, they need provide a clear definition and associated analysis criterion of what constitutes a theta sequence, perhaps using a peer prediction analysis along the lines of Harris et al. 2003 Nature; Foster and Wilson, 2007 Hippocampus; Chadwick et al., 2015 eLife. Judging by the authors' comments, it seems they are aware of this issue and don't in fact think that their results can be explained by independently phase precessing cells, but if this one of their central claims then a clear definition and associated analysis should be provided up front as well as throughout the relevant sections.

Claim 2: The authors need to be explicit about what, if any, claim they are making about theta sequences being trajectory-specific. If the claim is that HC theta sequences represent alternative futures before making a choice, i.e. recruiting cells representing positions after the choice point, this has been shown previously (Johnson and Redish, 2007 J Neurosci; Kay et al., 2020 Cell). If, on the other hand, the claim is that based on tuning curves prior to the choice point only, i.e. HC and/or PFC theta sequences are trajectory-specific ("splitter sequences") this would be novel, but requires analyses to determine whether the observed sequences are more split than would be expected from a resampled distribution.

Claim 3: This is potentially the most interesting finding, but as currently presented I found it confusing: isn't there an inconsistency between on the one hand, the claim that behavioral-timescale activity supports choice prediction based on pre-choice point (CP) activity in HC (Figure 1H), and on the other, the result that HC pre-CP theta sequences cannot predict choice (Figure 6H)? Is this because the decoding followed by binary classification (as L or R) of theta sequences ignores firing rate differences that could be used to decode future choice? Or is this because the theta sequence analysis throws out some part of the data that is included in the behavioral-timescale analysis? The concern is that if it is the former, a different classifier that uses firing rates would lead to a quite different result; if it is the latter, it seems the current result could be quite sensitive to arbitrary decisions of what data to include. The authors should discuss, and perhaps perform additional analyses, to explain the apparent inconsistency (see minor comments below for suggestions).

Claim 4: By itself, the current analyses of the HC/PFC interaction during SWR periods does not seem enough to conclude that there is something important about the interaction unless the authors also test how predictable performance is from HC and PFC content alone (perhaps with a GLM that could determine how much the prediction is improved by adding their measure of HC-PFC coordination). Also, how does this relate to the Shin et al. Neuron 2019 paper showing that CA1-PFC coordination could distinguish between replay of chosen and non-chosen path: is this just a restatement of that same result?

Regarding the overall interpretation that HC and PFC interact during memory guided decision making: wouldn't that require that HC and PFC theta sequences should be related? In fact, because PFC but not HC theta sequences predict upcoming choice, would that not be equally or more consistent with these structures not interacting? The analysis in Figure 6L could potentially speak to this point, but without additional analysis it is not clear to me if the reported result isn't just a consequence of PFC but not CA1 sequences being predictive of upcoming choice. Without a more direct demonstration of theta sequence coordination and a relationship to choice, the evidence for interaction seems to be mostly based on the SWR analysis, which as pointed out above needs to be more thorough in order to support an interaction.

– Different criteria for selecting splitter cells are used in HC and PFC, why not use the same criteria for both?

– Goal vs trajectory representation: "maintained representations of current goals"; "modulated by journeys and goals" - the authors should avoid making claims about goal representation: unless the authors analyze error trials, goal representation is confounded with memory of the previous trial. In addition, splitter cells seem to be better described as representing trajectories rather than goals (Grieves et al., eLife 2016).

– The replay results are given relatively high importance in the discussion and feature in the abstract, but are only presented in a supplementary figure; its relative prominence across sections should be made more consistent.

– "In this delayed alternation task, animals had to traverse a delay section ..." please clarify if an actual delay was enforced (i.e. keeping the animal confined to the central stem for some specified time) or if the task was self-paced. This is important because adding delays to continuous alternation tasks is known to make the tasks hippocampus-dependent (e.g. Ainge et al. Hippocampus 2007).

– Figure 1D: if the goal is to show a splitter cell, the authors should show an example where the signal splits in the stem, not around the choice point (where we know behavior is likely to differ)

– "approximately 50% of all theta sequences in both CA1 and PFC met these additional stringent criteria": please explain why the Figure 2—figure supplement 1 - panel D shows a different number (around 10%)?

– The authors should cite and discuss Stout and Griffin 2020 Frontiers Behavioral Neuroscience and Kinsky et al., 2020 Nature Communications as some of their findings are similar to those reported here.

– "both theta-associated mechanisms at the two timescales" - isn't there just one timescale for theta-associated mechanisms, i.e. theta sequences?

– "we have previously shown a relationship between forward and reverse CA1-PFC replay with past and future trajectories respectively (Shin et al., 2019)." It seems that in the cited paper forward replay was associated with future trajectories while reverse replay was associated with past ones, while the text implies the reverse - shouldn't it be "with future and past trajectories respectively"?

– "resampled the incorrect trials with replacement to match the correct trials" - shouldn't it be the reverse (downsampling correct trials?)

– "spatial fields" or "place fields" are generally used to refer to specific regions of high firing of a place cell (O'Keefe, 1976). Instead it seems that what the authors mean here is "rate maps", i.e., representations of firing rate as a function of space including all activity of the cell. Indeed, no criteria for place field detection are indicated in this paragraph. The authors could replace "spatial fields" by rate maps when appropriate and only use spatial fields or place fields when considering the actual place fields to avoid any confusion.

---

## [Author Response]

Essential Revisions:Reviewers agreed that this is a strong submission with beautiful data and figures, as well as analyses that are appropriate and solid. However, they felt that there was a lack of clarity regarding the paper's major conclusions and how some of the analyses led to those conclusions. Reviewers are confident that the authors will be able to address these concerns. No new data are required to support the conclusions. Please see the separate review below for details.

We thank the reviewers for their constructive comments and feedback, which have substantially improved the quality of our manuscript. The reviewers overall found our study important and results significant. Since the manuscript covers a lot of ground, the reviewers had suggestions to clarify the primary claims, some results and interpretations through additional explanation or analyses, which we have fully addressed in the revision.

Reviewer #1 (Recommendations for the authors (required)):1) To avoid confusion, the authors should perhaps reconsider using the phrase "population spikes" here. This phrase often has a different meaning in neurophysiology from what is meant here.

We thank the reviewer for this helpful suggestion. We have changed the phrase “population spikes” to “the firing pattern”.

2) To improve readability, the authors may want to consider not using the needless, non-standard abbreviation CP for choice point.

We appreciate the reviewer’s concern. We adopted the abbreviation CP for choice point from Singer et al., 2013. To avoid confusion, we have now used the full term, choice point, whenever possible (e.g., Figures 1 and 6, Figure 6—figure supplement 1, and Figure 7—figure supplement 1), except for a few figure panels, where the text becomes too crowded and hinders visualization. In these instances, we redefine the abbreviation for clarity.

3) In the figures, perhaps a different color than blue could be used for the left arm to improve clarity and readability. The color blue indicates outbound in some figures/panels and left arm trajectory in other figures/panels.

We thank the reviewer for this thoughtful comment. We apologize that the colors for left choice and outbound trajectories were difficult to discern, and have now used a different color to indicate left choice in all related figures (e.g., Figures 1 and 6).

4) I am confused by the broken y-axis in Figure 1B. Why is the axis disrupted with 0.4 written twice?

We apologize for not being clear on this. The top and bottom segments of the y-axis are segregated at 0.4 to emphasize the data in the relevant range, and 0.4 was repeated erroneously. For clarity, we have updated this representation.

5) The distance index in Figure 3C-D is not described in the Materials and methods section. The description in the figure legend is confusing. The description states that it compares posterior probabilities for future versus past locations, making it sound as though this means future and past relative to the present location of the animal. However, the figure legend also refers to theta phases, making it sound as though this measure was calculated based on theta phases of spikes. How exactly was this measure computed?

We apologize for not having provided enough detail about the distance index. To calculate the distance index of each candidate event, decoded probabilities ±60 cm around the animal’s location, and ±1⁄2 cycle around the 0 phase of the theta cycle (i.e., -π to 0 as 1^st^ half of theta phases, and 0 to π as 2^nd^ half of theta phases), were divided equally into four quadrants. The distance index of the 1^st^ half of theta phases is thus measured by comparing the probabilities in the quadrants ahead (quadrant III, future) and behind (quadrant II, past) as (III – II)/ (III + II). Similarly, the distance index of the 2^nd^ half of theta phases is thus measured as (IV – I)/ (IV + I) (Author response image 1).

**Author response image 1. sa2fig1:** 

We have now added these details into the Materials and methods section:“To calculate the distance index of each candidate event (Figure 3C and D), decoded probabilities ±60 cm around the animal’s current location, and ±1⁄2 cycle around the 0 phase of the theta cycle (i.e., -π to 0 as 1st half of theta phases, and 0 to π as 2nd half of theta phases), were divided equally into four quadrants (Feng et al., 2015). The distance index of the 1st half of theta phases is thus measured by comparing the probabilities in the quadrants ahead (quadrant III, future) and behind (quadrant II, past) as (III – II)/ (III + II). Similarly, the distance index of the 2nd half of theta phases is thus measured as (IV – I)/ (IV + I).”

6) The claim "Note the robust difference of CA1-PFC reactivation of taken versus not-taken path during correct trials compared to that during incorrect trials" is not well supported because no interaction effect is reported (i.e., interaction showing that CA1-PFC reactivation strength for same vs. alternative paths was significantly different for correct vs. incorrect trials). It seems as if only multiple paired t-tests were used here. Also, the data for correct and incorrect trials are plotted on different y-axis scales for panels A and B in Figure 7—figure supplement 2, making comparisons of effects difficult.

We thank the reviewer for raising this point. As per the suggestion, we have now removed the claim “Note the robust difference of CA1-PFC reactivation of taken versus not-taken path during correct trials compared to that during incorrect trials”. We have also changed the y-axis scale for panel A as the same for panel B in Figure 7—figure supplement 2.

The claim of “robust difference” for correct trial compared to incorrect trials referred to lower and significant *p*-values observed for correct trials compared to incorrect trials (correct trials: *p* = 0.0016 and 0.0037 for forward and reverse events, respectively; incorrect trials: *p* = 0.56 and 0.024 for forward and reverse events, respectively; Figure 7—figure supplement 2A-B). These *p*-values were derived from paired tests on a replay event-by-event basis. The rationale for the paired test here is that the strength of a replay event is affected by the number of active neurons and different aspects of selectivity of these neurons during the event. Since replay events during correct versus incorrect trials are two different sample sets, multiple paired tests and interaction effects between correct versus incorrect trials were thus not used.

To assess the trial-by-trial difference of replay strengths for correct and incorrect conditions, which may not be detectable and/or suitable using multiple paired tests, as the reviewer also noted, we therefore used SVM classifiers to examine whether the replay strengths have predictive power for correct and incorrect responses on a trial-by-trial basis. Indeed, significant differences between correct and incorrect trials are seen using this method, and we now report this result in the main figure (Figure 7F).

Reviewer #2 (Recommendations for the authors (required)):In this manuscript, Tang and colleagues extend previous work on medial prefrontal-hippocampal interactions in several important ways. In rats performing a well-validated spatial working memory task, they demonstrate the existence of theta sequences in PFC ensembles that mirror the organizational properties of theta sequences in hippocampus (albeit with less reliability than the canonical spatial representations in hippocampus). They show further that PFC theta sequences are coordinated with theta sequence representations in hippocampus; specifically, when hippocampus represents the animal's impending choice destination in a theta cycle, there is an increased likelihood that the simultaneous PFC representation shows the same representation. Finally, the authors show that the content of these fast timescale "cognitive" representations in theta cycles is dissociable from slower "behavioral timescale" representations. Specifically, at the behavioral time scale so called "splitter cells" provide sufficient information to decode impending choice from either hippocampus or PFC. In contrast, fast time scale representations diverged between the two structures before the animal committed to one choice option, with PFC representing the animals upcoming choice and hippocampus representing both possible choice options. This dissociation, however, was dynamically modulated, because after animals committed to a decision, both structure represented the chosen outcome.I think this manuscript is a potentially important contribution to our understanding of PFC-hippocampal dynamics. First, the identification of clear theta sequences in PFC is, on its own, an important result that is complementary to a recent report showing reactivation of spatially-tuned PFC cells that is highly reminiscent of reactivation of place cells in hippocampus (Kaefer...Csicsvari, 2020). In addition, however, the authors probe the coordination of prefrontal and hippocampal theta sequences, and find that there is an apparent relationship that is dynamically modulated by the behavioral demands of the decision making task. Together, I think the body of work reported here is an important contribution to our understanding of how these structures work together.The analysis is described clearly and with excellent detail, and appears rigorous and appropriate to the authors' questions. This is an incredibly information-rich manuscript, but it is well organized and well explained throughout. Overall, I think this is a strong contribution, and I have only very minor suggestions to improve it.At several points the authors mention that results in the present manuscript replicated their previous report (Shin et al 2019). This is important, but is somewhat tempered by the fact that 6/9 rats were reanalyzed from that 2019 data set.

We agree with the reviewer and have reworded ‘replicate’ to state ‘we have previously shown/ reported’, or that the results were consistent with our previous reports. This was especially relevant for the CA1-PFC reactivation results originally reported in Figure 7—figure supplement 2. Here, we would also like to note that the trial-by-trial prediction of correct and incorrect responses based on replay events (originally reported in Figure 7—figure supplement 2E-F) was a new result which was not achieved in Shin et al. 2019, and therefore not just a simple replication of our previous finding. Given the novelty and importance of the replay prediction result, especially in the context of lack of error prediction by theta sequences, we have now elevated this result to the main figure (Figures 7E and 7F).

Reviewer #3 (Recommendations for the authors (required)):This is an unusual paper in that it covers a large amount of ground, with 4-5 main claims that each individually have been (or could be) the subject of entire papers. In its attempt to integrate each of these findings into an overall theory, the paper reads a bit like a hybrid between a review and an empirical paper. One the one hand, this is a strength because it aspires to much-needed synthesis; on the other hand, I found it difficult to identify which statements are the primary empirical claims advanced in the paper, and which statements are interpretations. Moreover, for those statements that looked like main claims, I am not convinced that as presented, there is sufficient support. Thus, overall, I think the paper requires (a) more explicit statements about what the key primary claims are, and (b) additional analyses to better support those claims and/or better separation of result and interpretation.To expand on the above: as far as I can tell, the main novel claims seem to be:1) the prefrontal cortex (PFC) expresses theta sequences2) theta sequences in both the hippocampus (HC) and PFC are trajectory-specific ("splitter sequences")3) PFC, but not HC, theta sequence content predicts future choice4) joint HC-PFC activity during replay predicts error vs. correct trialsBased on these findings, the authors conclude that interaction at multiple timescales between HC and PFC supports memory-guided decision-making.If adequately supported, each of these claims would be a useful addition to the field, with the dissociation between HC and PFC theta sequences of particular interest. However, as currently presented, the claims seem out of step with the underlying evidence; in each case, this overall issue could be addressed by bolstering the analyses and/or by making the claims more precise or toned down.

We thank the reviewer for noting the importance of our work and providing many constructive suggestions to improve our manuscript.

Regarding primary new claims, the findings of prefrontal theta sequences, theta cycle skipping and theta sequence prediction are all novel, to the best of our knowledge. Further, we would like to argue that it is a significant strength of the manuscript that by presenting these findings in context of hippocampal phenomena, we are able to integrate both hippocampal and prefrontal results for behavioral sequences, theta sequences and replay sequences in a unified framework for memory-guided decision making. While we are aware of the extensive work on hippocampal sequences which is cited in the manuscript, there have been only very few studies investigating the relationship between these different sequences, even just in the hippocampus (e.g., Papale et al., 2016; Drieu et al., 2018), which is a major goal in the field (Pezzulo et al., Ann. N.Y. Acad. Sci., 2017). In the current study, we therefore present the new prefrontal results in the context of hippocampal sequences, and also build upon our previous replay results (Shin et al., 2019) to contrast the roles of the three types of sequences in memory-guided decision making. We hope the reviewer will share our vision and appreciate the strength of these findings.

We have now provided explicit statements and an updated focus on these primary advances of our current study in the Discussion section:

“In this work, we discovered theta sequences, theta cycle skipping and theta-sequence prediction of behavioral choices in PFC. These prefrontal phenomena follow a succession of results on hippocampal sequences, but our study points to different yet complementary roles of prefrontal and hippocampal sequences at multiple timescales. By dissecting fast cognitive-timescale sequences from slow behavioral-timescale sequences in a spatial working-memory and navigation task, these findings thus provide a unified framework that integrates hippocampal and prefrontal mechanisms of multi-timescale cell-assembly dynamics for memory-guided decision making.”

We have also followed the reviewer’s suggestions and provided additional evidence with new analyses and explanations to support our main novel conclusions. In overview, (1) we have now provided evidence that theta sequences in CA1 and PFC observed here, as ensemble-level phenomena, cannot be fully accounted for by single-neuronal phase precession, and thus the observation of the PFC theta sequences is a genuine and novel finding of the present work; (2) we have further clarified that our finding of “trajectory-specific” CA1 and PFC theta sequences is an important extension of the previously observed choice coding of theta sequences in the hippocampus; (3) we have demonstrated that the difference between behavioral- and theta-sequence coding of choices in CA1 and PFC is not a result of the methodological concerns suggested; (4) we have made changes in the presentation of the replay results, and added more explicit interpretation of the theta-sequence coordination result to enhance clarity. We believe that these changes, prompted by the reviewer’s suggestions, make the Results clearer and more accessible.

Specifically:Claim 1: What counts as a "theta sequence" requires clarification. Do the authors consider the ensemble pattern generated from independently phase precessing cells a theta sequence? If yes, then the claim that mPFC has theta sequences follows directly from previous single cell results (Jones and Wilson, 2005 PLoS Biology; Zielinski et al., 2019 J Neurosci) and they should acknowledge this. If no, they need provide a clear definition and associated analysis criterion of what constitutes a theta sequence, perhaps using a peer prediction analysis along the lines of Harris et al. 2003 Nature; Foster and Wilson 2007 Hippocampus; Chadwick et al. 2015 eLife. Judging by the authors' comments, it seems they are aware of this issue and don't in fact think that their results can be explained by independently phase precessing cells, but if this one of their central claims then a clear definition and associated analysis should be provided up front as well as throughout the relevant sections.

We thank the reviewer for this suggestion, and we agree that it is an important point, and investigating the relationship between theta sequences and phase precession could further strengthen our results. As the reviewer suggested, we have now performed the phase-jitter analysis from Foster and Wilson *Hippocampus* 2007 (Author response image 2), and as expected, theta phase precession by itself does not account for the full extent of observed theta sequences in CA1 and PFC.

**Author response image 2. sa2fig2:** Cumulative distributions of theta sequence scores (or correlations, *r*) for actual and phase-jittered events. (**A**) Distributions for events during outbound navigation (****p = 3.72e-96, and 1.28e-39 for CA1 and PFC, respectively, KS tests). (**B**) Distributions for events during inbound navigation (****p = 2.39e-132, and 4.29e-59 for CA1 and PFC, respectively, KS tests). Gray lines: phase-jittered events for a given brain region. Orange line: original CA1 events. Black lines: original PFC events.

As the reviewer noted, determining the relationship between theta sequences and phase precession in the hippocampus has been a topic of active experimental and computational work. This relationship, however, hasn’t been studied before in PFC and is thus of interest. To address this issue, we used the phase-jitter analysis from Foster and Wilson *Hippocampus* 2007. In brief, for any given cell at any given position, there is a range of theta phases at which the cell will fire. Therefore, within each candidate theta sequence event, for each spike, we randomly chose a phase from the distribution of possible phases, and shifted the spike time accordingly. The correlation between cell and time for each event was thus altered by the phase jittering (Foster and Wilson, *Hippocampus*, 2007). “If this (phase-position) relationship alone determines the occurrence of theta sequences, it should be possible to choose randomly from any of the phases available for a given cell and given position, and then observe theta sequences to *the same degree*” (Foster and Wilson, *Hippocampus*, 2007).

The response figure (Author response image 2) shows the distribution of correlations across 100 phase-jitter shuffles of all events (gray lines), compared with the distribution of correlations of original events (black and orange lines for PFC and CA1, respectively). Although the difference is more apparent for CA1 than PFC, correlations for actual events were significantly greater than those for shuffles in both regions (p(CA1) = 3.72e-96, and p(PFC) = 1.28e-39 for outbound; p(CA1) = 2.39e-132, and p(PFC) = 4.29e-59 for inbound; KS tests).

We have now included this result in a new figure supplement (Figure 2—figure supplement 2), and added the following description into the Results section:

“To test whether theta phase precession can account for the occurrence of theta sequences, we performed a phase-jitter analysis (Foster and Wilson, 2007), in which we randomly chose a phase of each spike from the distribution of possible phases of that cell in the position bin and shift the spike time accordingly for each candidate event. We found that sequence scores of actual events were significantly greater than those of shuffles in both CA1 and PFC (Figure 2—figure supplement 2), and thus theta phase precession does not account for the full extent of observed theta sequences in CA1 and PFC.”

Claim 2: The authors need to be explicit about what, if any, claim they are making about theta sequences being trajectory-specific. If the claim is that HC theta sequences represent alternative futures before making a choice, i.e. recruiting cells representing positions after the choice point, this has been shown previously (Johnson and Redish, 2007 J Neurosci; Kay et al., 2020 Cell). If, on the other hand, the claim is that based on tuning curves prior to the choice point only, i.e. HC and/or PFC theta sequences are trajectory-specific ("splitter sequences") this would be novel, but requires analyses to determine whether the observed sequences are more split than would be expected from a resampled distribution.

We thank the reviewer for the opportunity to clarify the trajectory-specific theta sequences. To determine the content of theta sequences, we used the trajectory-specific tuning curves at the behavioral timescale (i.e., rate maps) as templates of the Bayesian decoder, as we described in the Materials and methods section:

“A memoryless Bayesian decoder was built for different trajectory types (for outbound, C-to-L vs. C-to-R; for inbound, L-to-C vs. R-to-C) to estimate the probability of animals’ position given the observed spikes (Bayesian reconstruction; or posterior probability matrix):

P(X,Tr|spikes)=P(spikes|X,Tr)P(X,Tr)P(spikes), where *X* is the set of all linear positions on the track for different trajectory types (i.e., *Tr*; *Tr* ∈{*L*, *R*}, where *L* represents the *L*-side trajectory, *R* represents the *R*-side trajectory),”

Therefore, for the theta sequences that didn’t sweep beyond the choice point (e.g., Figure 6C, event 1), their content was determined by the tuning curves prior to the choice point only. On the other hand, for the theta sequences that swept beyond the choice point, the tuning curves after the choice point (or non-splitter) also contributed, similar to previous findings (Johnson and Redish, *JNeurosci*, 2007; Kay et al., *Cell*, 2020). Therefore, as the reviewer correctly pointed out, our findings extend these previous findings to show trajectory-specific theta sequences in CA1 and PFC. While previous reports show that splitter cells encode distinct trajectories at the behavioral timescale (e.g., Frank et al., 2000; Wood et al., 2000; Ito et al., 2015), our finding suggest that splitter information can also be utilized at fast theta timescales. However, we would like to note that, a recent study (Kay et al., *Cell*, 2020) has investigated the relationship between CA1 splitter cells and theta-timescale activity, and suggested that a splitter cell’s “firing on the non-preferred path is consistent with representation of a possible future, or hypothetical, location” at the theta timescale, although the decoding of theta sequences using splitter templates was not directly tested, which we have studied here. This previous study had also established theta-timescale representation of real and hypothetical heading directions in the hippocampus, which is similar to our trajectory-specific theta sequences in principle.

To determine whether the observed sequences are significantly more split for different choices, the observation of each sequence was tested against two resampled distributions, in which the trajectory coding information (i.e., spatial shuffling) and cell identities (i.e., cell ID shuffling) were randomly shuffled. This method is also similar to that used in previous studies to determine whether the observed theta sequences are significantly more split for different heading directions (e.g., Wang et al., *Science*, 2020; Kay et al., *Cell*, 2020).

We now discuss this point more thoroughly in the manuscript. Specifically, we have included the following in the Discussion section:

“These findings build on previous results that demonstrated theta-timescale activity patterns in the hippocampus representing future locations by recruiting cells encoding positions after the choice point (i.e., non-splitters; Johnson and Redish, 2007; Kay et al., 2020; Papale et al., 2016), and show that trajectory-specific coding of splitter cells at the behavioral timescale is preserved in fast theta sequences for representing future choices prior to the choice point as well.”

Claim 3: This is potentially the most interesting finding, but as currently presented I found it confusing: isn't there an inconsistency between on the one hand, the claim that behavioral-timescale activity supports choice prediction based on pre-choice point (CP) activity in HC (Figure 1H), and on the other, the result that HC pre-CP theta sequences cannot predict choice (Figure 6H)? Is this because the decoding followed by binary classification (as L or R) of theta sequences ignores firing rate differences that could be used to decode future choice? Or is this because the theta sequence analysis throws out some part of the data that is included in the behavioral-timescale analysis? The concern is that if it is the former, a different classifier that uses firing rates would lead to a quite different result; if it is the latter, it seems the current result could be quite sensitive to arbitrary decisions of what data to include. The authors should discuss, and perhaps perform additional analyses, to explain the apparent inconsistency (see minor comments below for suggestions).

We thank the reviewer for pointing this out. The reviewer raised the concern that the difference between behavioral versus theta sequences that we found in HC could potentially be accounted for by the fact that we used different decoders for theta versus behavioral sequences, of which theta-sequence decoder ignored firing rate differences, but only used the sequential structure. First of all, we would like to note that our results of hippocampal behavioral and theta sequences confirm previous reports that HC theta sequences on the center stem that sweep beyond the choice point serially encode alternatives (see also responses to Comment 2; e.g., Johnson and Redish, 2007; Kay et al., 2020), and HC trajectory-dependent activity at the behavioral timescale consistently represents upcoming choice on the center stem (e.g., Takahashi, *eLife*, 2013; Ito et al., 2015). Furthermore, we would like to point out that it is not surprising that decoding of the same data at different timescales can reveal different information (e.g., Kayser et al., *Neuron*, 2009; Huxter et al., *Nature*, 2003; Rich and Wallis, 2016; Zheng et al., 2020). To directly address this concern, we have now performed a decoding analysis for behavioral sequences that is identical to theta sequence decoding. This result shows that the difference of behavioral versus theta sequences was not a simple result of different decoders used.

First, the reviewer is correct in pointing out that the Bayesian decoders for behavioral-timescale activity and theta sequences are indeed not identical. For behavioral-timescale activity, we used a bin-by-bin decoding that takes “firing rate differences” into account: for a behavioral sequence before the choice point that can be divided into *N* behavioral-timescale bins (bin = 120 ms), of each time bin *i* (*i* ∈[1, *N*]), the location (*x_i_*) and choice (i.e., trajectory type, *tr_i_*) with the maximum decoded probability (i.e., max(p(*x_i_*,*tr_i_*|**spikes**))) was compared to the actual position and choice of the animal in that bin. On the other hand, for a theta sequence that can be divided into *N* theta-timescale bins (*S* = [*s_1_*, *s_2_*, …, *s_N_*], *s_i_* is the firing pattern in the *i*-th bin; bin = 20 ms), the posterior probability matrix **Pmat** (*p*(*X*,*Tr*|**spikes**), *X* = [*x_1_*, *x_2_*, …, *x_N_*]) was used to compute a trajectory score for each trajectory type (i.e., *Tr*, *Tr* ∈{*L*, *R*}; see Materials and methods), and compare with its shuffled distributions, with the significant trajectory type that has the highest sequence score as the decoded trajectory. Therefore, this decoder emphasized the sequential structure of a theta-sequence event, instead of just firing rate differences between the two trajectory types. We would like to note that we used the bin-by-bin decoder of behavioral sequences to compare the decoding accuracy with previously reported results of behavioral sequences (e.g., Ito et al., *Nature*, 2015; Takahashi, *eLife*, 2013; Harvey et al., *Nature*, 2012). For theta sequences, however, a “sequential” decoder was often employed (Feng et al., *JNeurosci*, 2015; Wang et al, *Science*, 2020; Zheng et al., *Neuron*, 2015).

However, we understand the reviewer’s concern about the differences of the decoders. There are two ways to make these two decoders identical: using a bin-by-bin decoder of maximum decoded probability for theta-timescale activity; and using a sequential decoder for behavioral sequences.

For the first approach, previous studies have used a bin-by-bin decoder for theta-timescale activity, just as the behavioral-sequence decoder, and found similar results that hippocampal theta-timescale activity can serially represent actual and alternative choices (Johnson and Redish, 2007; Papale et al., 2016).

**Author response image 3. sa2fig3:** Behavioral-sequence representations of choices in CA1 and PFC. (**A** and **B**) Single-event examples of (**A**) CA1 and (**B**) PFC behavioral sequences simultaneously recorded during a C-to-R outbound trial. Left: corresponding linearized spatial firing rate maps (blue colormap for L-side trajectory, red colormap for R-side trajectory). Right: Bayesian decoding with sequence score (*r*) and p-values based on shuffled data denoted. Cyan lines: actual position. Note that a few single-bin decoding errors of current choices occurred along a sequence. Also note the global remapping of some cells on different trajectory types (CP: choice point). (**C** and **D**) Percent of behavioral sequences representing actual or alternative choice before the choice point (CP) for outbound (OUT) and inbound (IN) trials in (**C**) CA1 and (**D**) PFC (****p < 1e-4, session-by-session rank-sum paired tests). Error bars: SEMs. Data are presented as in Figure 6G. The choice representation of each behavioral sequence was determined by a sequential decoder, similar to that for theta sequences.

On the other hand, given that we focused on the “sequential” structure of ensemble activity in this study, we have now used a sequential decoder for behavioral sequences, just as the theta-sequence decoder (Author response image 3). To do so, we took "an event" of behavioral sequences, as the firing pattern during a single trial (trajectory) on the center stem, and used it as an input to the Bayesian decoder with spatial shuffling, just as used in theta-sequence decoding, but with time window of 400 ms (instead of 20 ms for theta sequences). Unlike CA1 theta sequences that serially encode actual and alternative choices, CA1 behavioral sequences preferentially encode the actual choice. Therefore, this result is consistent with the one found by the bin-by-bin decoder (Figure 1), suggesting that the difference between behavioral versus theta sequences cannot simply be accounted for by the choice of the decoders. Note that the more robust representations here, compared to bin-by-bin decoding shown in Figure 1, are expected, because this measure took the whole sequence structure (i.e., *S* = [*s_1_*, *s_2_*, …, *s_N_*]) into account, which is less noisy than using the information within a single bin only (i.e., *s_i_*). Also, we note that these results are consistent with previous findings that splitters cells show not only rate remapping (firing rate difference) but also global remapping (firing position difference) on different trajectory types (e.g., Wood et al., 2000; Frank et al., 2000; Fujisawa et al., 2008; examples in panels **A** and **B** of Author response image 3).

Regarding the latter concern, i.e. whether the observed difference between behavioral and theta sequences is because that the theta sequence analysis throws out some part of the data that is included in the behavioral-timescale analysis, first, we show above that using the same classifier for the distinct timescale sequences still yields the same result. Further, as noted above, it is not surprising that decoding of the same data at different timescales can reveal different information (e.g., Kayser et al., *Neuron*, 2009; Huxter et al., *Nature*, 2003; Rich and Wallis, 2016; Kay et al., 2020, Zheng et al., 2020). We also want to note that the proportion of significant theta sequences included in this analysis was determined by the Bayesian decoder with shuffling procedures, with numbers consistent with many previous reports of theta sequences in the hippocampus (Gupta et al., 2012; Drieu et al., 2018; Schmidt et al., 2019), and we did NOT make any arbitrary decisions of what data to include. Finally, we also show that applying additional criteria for significance to the Bayesian decoder, which changed the number of theta sequences used, also revealed the same result (Figure 6—figure supplement 2A, left), arguing against the possibility of sensitivity to data used.

To avoid potential confusion, we have now underscored the different window sizes for decoding in the Results section:

(for behavioral sequences) “To directly assess the cell-assembly representation of the animals’ choices, we used a memoryless Bayesian decoding algorithm (see Materials and methods; decoding bin = 120 ms) (Shin et al., 2019).”

and:

(for theta sequences) “In order to statistically identify theta sequences in CA1 and PFC, the firing pattern within each candidate hippocampal theta cycle (≥ 5 cells active in a given brain region) during active running were analyzed using the Bayesian decoding approach (decoding bin = 20 ms),…”

Claim 4: By itself, the current analyses of the HC/PFC interaction during SWR periods does not seem enough to conclude that there is something important about the interaction unless the authors also test how predictable performance is from HC and PFC content alone (perhaps with a GLM that could determine how much the prediction is improved by adding their measure of HC-PFC coordination). Also, how does this relate to the Shin et al. Neuron 2019 paper showing that CA1-PFC coordination could distinguish between replay of chosen and non-chosen path: is this just a restatement of that same result?

We appreciate the reviewer’s concern. We have investigated hippocampal-prefrontal interactions during SWRs in detail elsewhere in several previous studies (Jadhav et al., 2016; Tang et al., 2017; Shin et al. 2019). In one of these studies (Jadhav et al., 2016), we have used GLMs to predict SWR-related spiking of PFC neurons from the activity of the simultaneously recorded CA1 ensembles, and “these findings demonstrate that there is close coordination between CA1 and PFC activity during SWRs”. Consistent results have been shown by several other subsequent studies (e.g., Wang and Ikemoto, *JNeurosci*, 2016; Yu et al., *eLife*, 2017).

For the replay result shown here, it is indeed consistent with our previous finding that CA1-PFC coordination could distinguish between replay of chosen and non-chosen paths (Shin et al., 2019), as the reviewer noted. Here, with additional data, we perform a trial-by-trial prediction of correct and incorrect responses based on replay events (Figure 7—figure supplement 2E-F in the previous version; now elevated to Figure 7F in the revised version), which is a new result and was not achieved in Shin et al. 2019. Finally, given the lack of error prediction by theta and behavioral sequences, the result of replay prediction thus provides further insights into the mechanisms underlying accuracy of memory-guided decision making. Together, these findings offer a consolidated view of multi-timescale sequences in the hippocampus and prefrontal cortex.

Regarding the overall interpretation that HC and PFC interact during memory guided decision making: wouldn't that require that HC and PFC theta sequences should be related? In fact, because PFC but not HC theta sequences predict upcoming choice, would that not be equally or more consistent with these structures not interacting? The analysis in Figure 6L could potentially speak to this point, but without additional analysis it is not clear to me if the reported result isn't just a consequence of PFC but not CA1 sequences being predictive of upcoming choice. Without a more direct demonstration of theta sequence coordination and a relationship to choice, the evidence for interaction seems to be mostly based on the SWR analysis, which as pointed out above needs to be more thorough in order to support an interaction.

The reviewer is correct that PFC theta sequences preferentially encode the actual choice, and PFC but not hippocampal theta sequences predict upcoming choice. It is common that two (even directly) interacting brain regions can encode different task information, which by itself cannot reveal whether these structures are interacting or not. For the CA1-PFC coordination in Figure 6L, we argue that this result cannot be simply accounted for by the fact that the majority of PFC theta sequences depict the actual choice, assuming independence of CA1 and PFC theta sequences. For synchronous CA1 and PFC theta sequences, if the representations of CA1 and PFC theta sequences (denoted as *Seq_CA1_* and *Seq_PFC_*, respectively) are stochastically independent, then *p*(*Seq_PFC_* = actual| *Seq_CA1_*= actual) = *p*(*Seq_PFC_* = actual| *Seq_CA1_*= alternative) = *p*(*Seq_PFC_* = actual) = 1- *p*(*Seq_PFC_* = alternative). Given that *p*(*Seq_PFC_* = actual) and *p*(*Seq_PFC_* = alternative) are significantly different from the chance level (50%; p’s < 1e-4, signed-rank tests compared to 50%), stochastic independence would predict that the distributions of sequences coherently representing actual and alternative (i.e., *p*(*Seq_PFC_* = actual| *Seq_CA1_* = actual) and *p*(*Seq_PFC_* = alternative | *Seq_CA1_*= alternative)) both significantly differ from the chance level. However, if they are dependent, different distributions for actual and alternative should be observed, which is indeed the case as shown in Figure 6L, i.e., *p*(*Seq_PFC_* = actual| *Seq_CA1_*= actual) significantly differs from the chance level (p’s = 0.0312), but *p*(*Seq_PFC_* = alternative | *Seq*_*CA1*_= alternative) does not (p = 0.94 and 0.48 for outbound and inbound, respectively; signed-rank tests).

In addition, we would like to note that the idea of CA1-PFC interactions at the theta timescale is also supported by previous evidence showing that PFC cells phase-lock and phase-precess to hippocampal theta oscillations (Jadhav et al., 2016; Jones and Wilson, 2005a; Siapas et al., 2005; Sigurdsson et al., 2010), as well as theta-frequency synchrony and coherent spatial coding between the hippocampus and PFC (Benchenane et al., 2010; Gordon, 2011; Hasz and Redish, 2020; Jones and Wilson, 2005b; Sigurdsson et al., 2010; Zielinski et al., 2019). Furthermore, disruption of prefrontal activity results in impaired theta sequences in the hippocampus (Schmidt et al., 2019). We have discussed these studies in our manuscript.

We have now explicitly mentioned and discussed this issue in the Results section:

“Note that this result cannot be simply accounted for by the fact that the majority of PFC theta sequences depict the actual choice, assuming independence of CA1 and PFC theta sequences (see Materials and methods).”

and in the Materials and methods section:

“For synchronous CA1 and PFC theta sequences, we examined if their representations were coherent or independent (Figures 6J-6L). […] However, if they are dependent, different distributions for actual and alternative should be observed (Figure 6L).”

– Different criteria for selecting splitter cells are used in HC and PFC, why not use the same criteria for both?

These criteria were defined arbitrarily based on the fact that |*SI*| = 0.384 ± 0.004 and 0.154 ± 0.003 for all CA1 and PFC cells recorded, respectively. We used these criteria to show examples of trajectory specific ensemble sequences in Figures 1F and 1G (the decoding and statistics of behavioral sequences were nevertheless performed using all cells recorded in a given region). We have used more stringent and conservative criteria with trial-label shuffling procedures to define trajectory-dependent cells (i.e., splitter cells; the trial labels (left or right) were randomly shuffled 1,000 times, and cells with a *SI* significantly higher than its shuffle surrogates (p < 0.05) were defined as splitter or trajectory-dependent cells), and reported statistics based on these shuffles in Shin et al., 2019.

– Goal vs trajectory representation: "maintained representations of current goals"; "modulated by journeys and goals" - the authors should avoid making claims about goal representation: unless the authors analyze error trials, goal representation is confounded with memory of the previous trial. In addition, splitter cells seem to be better described as representing trajectories rather than goals (Grieves et al., eLife 2016).

We agree with the reviewer about careful wording for “goal representation”. To better reflect our findings, “maintained representations of current goals” was rephrased to “maintained representations of current choices”, and “modulated by current journeys and goals” was changed to “modulated by current journeys”.

– The replay results are given relatively high importance in the discussion and feature in the abstract, but are only presented in a supplementary figure; its relative prominence across sections should be made more consistent.

Considering the novelty and importance of the replay results as we discussed in Comment 4 (1. the trial by-trial prediction of correct and incorrect responses based on replay events is a new result that wasn’t able to be achieved in Shin et al., 2019; 2. this result, together with similar properties of behavioral and theta sequences during correct and incorrect trials, provides further insights into the mechanisms underlying correct and incorrect responses, as well as a consolidated view of multi-timescale sequences in the hippocampus and prefrontal cortex), we have now elevated the replay prediction result into the main figure (Figures 7E and 7F). We have one sentence in the Abstract describing this result, i.e., “whereas replay sequences during inter-trial periods were impaired prior to navigation”.

– "In this delayed alternation task, animals had to traverse a delay section ..." please clarify if an actual delay was enforced (i.e. keeping the animal confined to the central stem for some specified time) or if the task was self-paced. This is important because adding delays to continuous alternation tasks is known to make the tasks hippocampus-dependent (e.g. Ainge et al., Hippocampus 2007).

The reviewer is correct that we did not enforce a delay in our task (i.e. keeping the animal confined to the central stem for some specified time). We have further clarified this point in the Results section:

“In this delayed alternation task, animals had to traverse a spatial delay section (i.e., common “center stem”; no enforced delay) of a W-maze on each trial.”

As for hippocampus-dependence of the task, we think that the reviewer is referring to tasks similar to maze-based continuous alternation without delay (e.g., continuous alternation on a Figure-8 or T-maze; Ainge et al., 2007). While the reviewer is correct that these tasks have been shown to be hippocampus independent, we would like to clarify a distinction with the W-maze alternation task. It has been shown in previous studies using lesions or optogenetic manipulations that animals are likely to accomplish such a continuous (T-maze/8-maze) spatial alternation task as a “habit” (Ainge et al., *Hippocampus*, 2007; Sabariego et al., *Neuron*, 2019), and such tasks do not depend on the hippocampus (Ainge et al., *Hippocampus*, 2007; Sabariego et al., *Neuron*, 2019), medial entorhinal cortex (Sabariego et al., *Neuron*, 2019), or prefrontal-hippocampal communication via nucleus reuniens (Ito et al., 2015). However, the sensitivity of the W-maze task to hippocampal and prefrontal functions is quite different from these continuous alternation tasks. It has been shown that rats with hippocampal lesions exhibited impaired learning of both inbound and outbound components of the task (Kim and Frank, 2009); disruption of SWRs can impair the learning of the outbound component of the task (Jadhav et al., 2012); contralateral hippocampal-PFC inactivation also impairs learning of the outbound component of the task (Maharjan et al., 2018). A distinguishing feature of the W-maze task is that working-memory (outbound) and reference-memory (inbound) demands are regularly interleaved between trials, and the time in the center arm and well serves as a built-in ‘‘delay’’ period (Kim and Frank, 2009), leading to the hippocampal dependence and the difference in learning this task compared to simple continuous alternation mazes.

– Figure 1D: if the goal is to show a splitter cell, the authors should show an example where the signal splits in the stem, not around the choice point (where we know behavior is likely to differ)

An example where the signal splits in the stem has been shown in Figure 1E. Given that both the center-stem tail of a spatial field with its peak located at the side arm or around the choice point, and a spatial field with its peak located within the center-stem could be used to define splitter cells and contribute to decoding of upcoming choices (e.g., Wood et al. 2000; Ito et al., 2015), we picked an example for each of these situations in Figures 1D and 1E, respectively.

– "approximately 50% of all theta sequences in both CA1 and PFC met these additional stringent criteria": please explain why the Figure 2—figure supplement 1 - panel D shows a different number (around 10%)?

We apologize for not being clear about this. “Approximately 50% of all theta sequences in both CA1 and PFC met these *additional* stringent criteria” refers to the fact that 50% of all theta sequences detected using spatial shuffles alone (Figure 2G; i.e., ~20% are significant) were determined as significant using both spatial and cell-ID shuffles (Figure 2—figure supplement 1D; i.e., ~10% are significant, which is ~50% of ~20%). We have now clarified this point in the legends of Figure 2—figure supplement 1D.

– The authors should cite and discuss Stout and Griffin 2020 Frontiers Behavioral Neuroscience and Kinsky et al., 2020 Nature Communications as some of their findings are similar to those reported here.

We thank the reviewer for alerting us of these references, which we have now included in our revised manuscript.

– "both theta-associated mechanisms at the two timescales" - isn't there just one timescale for theta-associated mechanisms, i.e. theta sequences?

“Both theta-associated mechanisms at the two timescales” was referred to the behavioral-timescale and theta-timescale mechanisms during locomotor periods, because both rate-map activity and theta sequences have been shown to be modulated by theta oscillations. We have further clarified this point as:

“Theta-associated mechanisms at both behavioral and theta timescales during active navigation supported retention during working memory periods, but did not predict errors in decisions.”

– "we have previously shown a relationship between forward and reverse CA1-PFC replay with past and future trajectories respectively (Shin et al., 2019)." It seems that in the cited paper forward replay was associated with future trajectories while reverse replay was associated with past ones, while the text implies the reverse - shouldn't it be "with future and past trajectories respectively"?

We thank the reviewer for bringing this to our attention and we apologize for this typo. We have now corrected this:

“We have previously shown a relationship between reverse and forward CA1-PFC replay with past and future trajectories respectively (Shin et al., 2019).”

– "resampled the incorrect trials with replacement to match the correct trials" - shouldn't it be the reverse (downsampling correct trials?)

It is correct that we used up-sampling (i.e., randomly sample, with replacement, the minority class to be the same size as the majority class), instead of down-sampling, for the ROC analysis. While both up-sampling and down-sampling are commonly used to resolve class imbalance for classification, downsampling always results in a loss of information, and the cost of misclassification of the minority samples is often higher than that of the majority samples. For these reasons, we chose up-sampling in this analysis.

– "spatial fields" or "place fields" are generally used to refer to specific regions of high firing of a place cell (O'Keefe, 1976). Instead it seems that what the authors mean here is "rate maps", i.e., representations of firing rate as a function of space including all activity of the cell. Indeed, no criteria for place field detection are indicated in this paragraph. The authors could replace "spatial fields" by rate maps when appropriate and only use spatial fields or place fields when considering the actual place fields to avoid any confusion.

We agree with the reviewer that “rate maps” is a more accurate description than “spatial fields” in this situation. We have now replaced “spatial fields” with “rate maps” (or “spatial firing rate maps”), whenever appropriate, in our revised manuscript. For a few analyses, in which specific regions of high firing were clearly defined (e.g., in *Spatial field asymmetry* section of the Materials and methods), “spatial field” was used, and it is consistent with recent finding that spatially localized fields across multiple neocortical regions depend on an intact hippocampus (Esteves et al., *JNeurosci*, 2021).